# Detecting and phasing minor single-nucleotide variants from long-read sequencing data

Zhixing Feng [1,2✉], Jose C. Clemente [1,2], Brandon Wong [3] & Eric E. Schadt [1,2,4]

Cellular genetic heterogeneity is common in many biological conditions including cancer, microbiome, and co-infection of multiple pathogens. Detecting and phasing minor variants play an instrumental role in deciphering cellular genetic heterogeneity, but they are still difficult tasks because of technological limitations. Recently, long-read sequencing technologies, including those by Pacific Biosciences and Oxford Nanopore, provide an opportunity to tackle these challenges. However, high error rates make it difficult to take full advantage of these technologies. To fill this gap, we introduce iGDA, an open-source tool that can accurately detect and phase minor single-nucleotide variants (SNVs), whose frequencies are as low as 0.2%, from raw long-read sequencing data. We also demonstrate that iGDA can accurately reconstruct haplotypes in closely related strains of the same species (divergence ≥0.011%) from long-read metagenomic data.

[1] Icahn Institute for Data Science and Genomic Technology, Icahn School of Medicine at Mount Sinai, New York, NY, USA. [2] Department of Genetics and Genomic Sciences, Icahn School of Medicine at Mount Sinai, New York, NY, USA. [3] Department of Biomedical Engineering, Johns Hopkins University, Baltimore, MD, USA. [4] Sema4, Stamford, CT, USA. ✉email: zhixing.feng@mssm.edu

Cellular genetic heterogeneity is prevalent in multiple biological conditions. For example, the microbiome contains multiple bacterial species with distinct genomes, and patients with infections may carry multiple bacterial strains. Likewise, in cancer, tumors are typically characterized by multiple cell types and cell lineages with different genomes. Deconvoluting such complex cellular genetic heterogeneity is critical to basic biology and precision medicine. Minor variants, which are defined as the variants with frequencies lower than 10% in a cell population, play a central role in deciphering cellular genetic heterogeneity. Short-read genome sequencing can effectively characterize a large number of cells simultaneously but cannot phase minor variants directly due to the limitation of read length, which is generally under 300 bp[1]. Long-read sequencing, on the other hand, can be used to overcome this limitation. The latest long-read sequencing technologies, including those by Pacific Biosciences (PacBio) and Oxford Nanopore (ONT), enable sequencing more than 100 billion bases in a single run and yield reads with lengths that can exceed 10 kb[2–4]. These advantages make it feasible to adopt long-read sequencing to study cellular genetic heterogeneity in the microbiome, bacterial co-infection, and cancer in finer details. Because of its long read length and high throughput, long-read sequencing has the potential to be applied to detect and phase minor variants at the single-molecule level without amplification. However, the error rate of raw long-read sequencing data is usually higher than 10%[1,3], and makes it difficult to detect variants whose frequency is lower than the sequencing error rate.

Most of the existing methods to detect minor SNVs are based on short-read sequencing data[5–14]. The vast majority of these methods scan the reference genome and detect SNVs or other variants locus-by-locus. These methods cannot be used for long-read sequencing data because they are based on the error pattern of short-read sequencing data, which is different from long-read sequencing data. Researchers have also tried to leverage the information of multiple SNVs to increase detection accuracy. V-Phaser and V-Phaser2[15,16], which were designed for short-read sequencing data, use the joint probability of SNV pairs to detect SNVs. However, to avoid combinatorial explosion, they only use the joint probability of two SNVs. We will discuss the limitations of such a restriction for long-read sequencing and demonstrate how it leads to false negatives in Results.

There are several methods designed specifically to detect variants from long-read sequencing data. The GenomicConsensus module (https://github.com/PacificBiosciences/GenomicConsensus) developed by PacBio generates a consensus sequence from the aligned PacBio reads and compares it to the reference genome to identify variants. Nanopolish[17] is a variant caller designed specifically for ONT data, and Clairvoyante[18] is a deep-learning-based tool for Illumina, PacBio, and ONT data. These methods assume that samples only have one or two haplotypes and therefore cannot be applied to detect minor variants. MinorSeq (https://github.com/PacificBiosciences/minorseq), developed by PacBio, is designed to detect minor variants but requires its input to be circular consensus sequencing (CCS) reads[19]. CCS is a special protocol of PacBio sequencing, which sequences each DNA molecule multiple times to increase accuracy. Recently, several tools have been developed to detect variants by leveraging haplotype information from long-read sequencing data[20–22], but they assume that the number of haplotypes is known. Thus, they cannot be applied to detect and phase minor variants.

There are several short-read-based methods available to phase minor SNVs[23–29]. These methods cluster the reads locally and phase distant SNVs, whose distances are longer than read length, using statistical models with strong assumptions. The major limitation of these methods is that they phase distant minor SNVs only based on indirect evidence because the read length is too short to span over the distant SNVs. This limitation can be overcome by using long-read sequencing data. The existing haplotyping methods for long-read sequencing data[20–22] assume there are only one or two haplotypes, and thus cannot be used to phase minor SNVs because the number of haplotypes is unknown.

In this work, we develop a tool named iGDA (in vivo Genome Diversity Analyzer) to address the challenges of detecting and phasing minor SNVs. iGDA can accurately detect and phase minor SNVs, whose frequencies are as low as 0.2%, in our testing data. To detect minor SNVs, iGDA leverages the information of multiple loci without restricting the number of dependent loci, and uses our proposed algorithm, Random Subspace Maximization (RSM), to overcome the issue of combinatorial explosion. To phase minor SNVs, iGDA uses our proposed algorithm, Adaptive-Nearest Neighbor clustering (ANN), which makes no assumption about a number of haplotypes. To evaluate the performance of iGDA, we test it on four pooled long-read sequencing datasets. The number of samples pooled in each dataset ranges from 65 to 755. The results demonstrate that iGDA can detect 85.8% to 96.7% of the real SNVs in these datasets at a false discovery rate (FDR) lower than 1%. Finally, iGDA can phase minor SNVs at average accuracies ranging from 90.7% to 98.7%. We also test iGDA on a pooled long-read metagenomic dataset consisting of 11 *Borrelia burgdorferi* strains and 744 other bacterial species, and discover that the accuracy of iGDA is sufficient to reconstruct haplotypes in closely related conspecific strains (strains belonging to the same species) only using one reference genome. The divergences between the distinguishable conspecific strains are as low as 0.011%. These results shed light on tackling a number of challenges such as extracting strain-resolved genome sequences from long-read metagenomic data and identifying multiple strains in co-infection.

## Results

**Detecting minor SNVs by leveraging information of multiple loci.** The major challenge of detecting minor SNVs is to distinguish between real SNVs and sequencing errors. It is especially difficult for raw data of long-read sequencing technologies, including those by PacBio and ONT, because they have relatively high error rates. However, we could leverage the fact that long reads can cover multiple SNVs to substantially increase detection accuracy. Intuitively, assuming that sequencing errors are independent, the same combination of sequencing errors at multiple loci is unlikely to repeatedly occur together on multiple reads. For example, in a pooled PacBio sequencing dataset consisting of 186 *Bordetella* spp. samples (Fig. 1a), the substitutions from the five marked loci occur together on 28 reads and there are 23,432 reads covering these five loci. The observed joint probability that these five substitutions occur together on the same read is $28/23{,}432 = 0.00119$, while the expected joint probability is less than $0.1^5 = 0.00001$ because the substitution error rate of raw PacBio reads is less than 0.1 on this dataset (Fig. 1b). The observed joint probability is over 100 times higher than the expected joint probability, so it is very likely that some of the five substitutions are real SNVs. However, the substitution rates of these five SNVs are 0.00569, 0.00845, 0.00748, 0.00960, and 0.00915 respectively and it is difficult to distinguish them from sequencing errors only based on the substitution rate (Fig. 1b). Based on these observations, we propose a framework that uses the conditional substitution rate instead of the substitution rate to detect SNVs. In this framework, for each substitution, we adopt the maximal

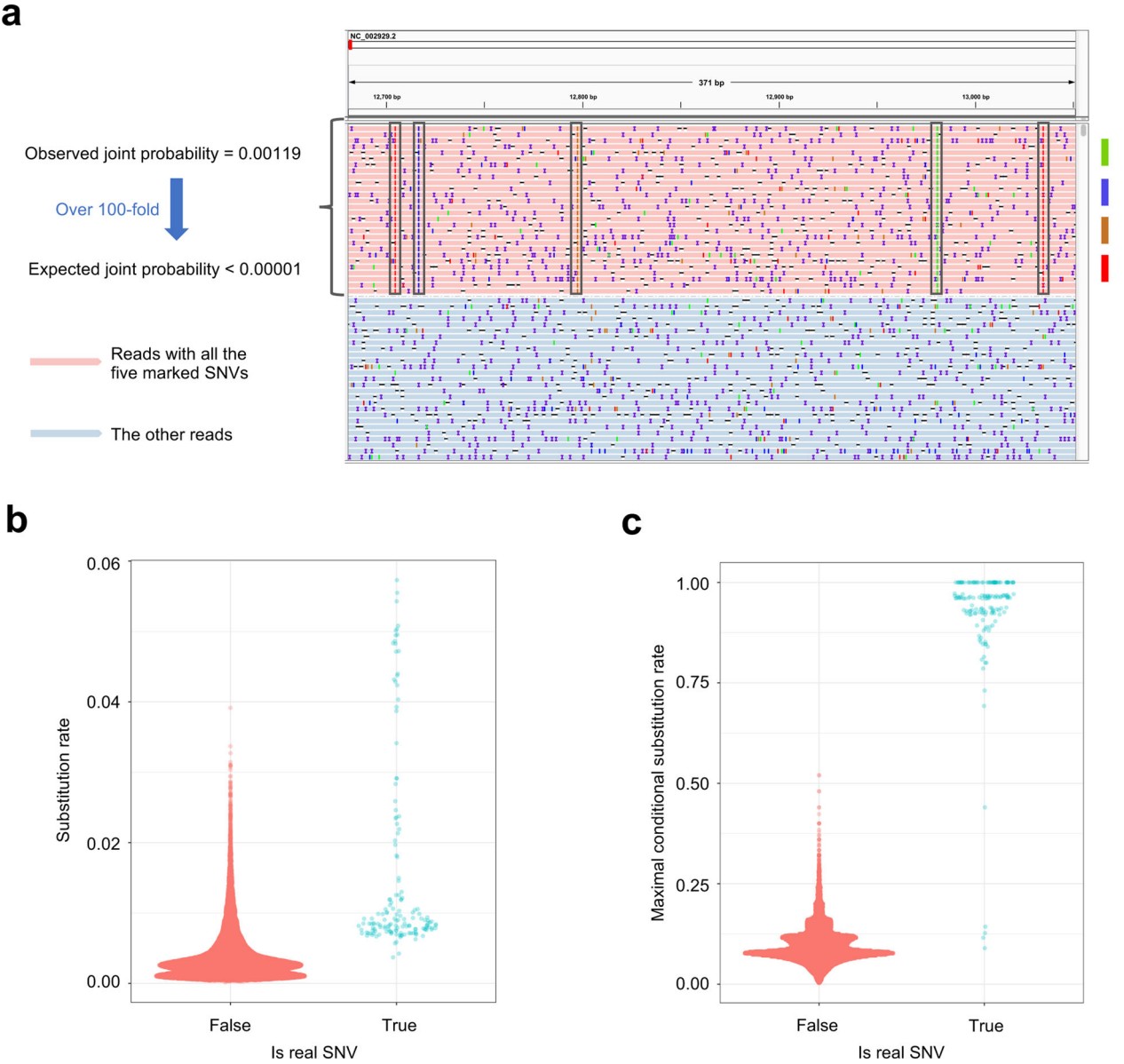

**Fig. 1 SNVs are dependent on each other. a** An IGV (Integrative Genomics Viewer)[34] snapshot demonstrating how to use the information of multiple loci to increase the detection accuracy of SNVs. The number of reads containing the five SNVs marked by black boxes is 28 and the number of reads covering the five SNVs is 23,432. The observed and expected joint probabilities of the five SNVs are shown to the left of the IGV snapshot. Some reads are not shown in the figure due to the limit of figure size. **b** The distribution of substitution rate on the *Bordetella* spp. data. No outlier is removed in the Sina plot. **c** The distribution of maximal conditional substitution rate estimated by the RSM algorithm on the *Bordetella* spp. data. No outlier is removed in the Sina plot. Source data are provided as a Source Data file.

probability of observing the substitution conditional on observing substitutions at $p$ other loci, defined as maximal conditional substitution rate, to detect whether the substitution is a real SNV. We call these $p$ loci "dependent loci". However, as the $p$ dependent loci are unknown, it is infeasible to enumerate all combinations of these $p$ loci to calculate the maximal conditional substitution rate due to the high computational cost. As $p$ is unknown, the number of combinations is about $\sum_{p=1}^{2l} C_{2l}^{p} = 2^{2l} - 1$ for each locus if the average read length is $l$. We propose an algorithm called RSM to estimate the maximal conditional substitution rate efficiently (Fig. 2a–c) (details are in "Methods"). As shown in Fig. 1c, on the *Bordetella* spp. data, the real SNVs and the sequencing errors are highly distinguishable based on the maximal conditional substitution rate calculated by the RSM algorithm.

It is very important to note that the number of dependent loci $p$ should not be fixed. Supplementary Fig. 1 shows an example that fixing $p$ can induce false negatives. In this example, the substitution at the locus 1 is independent with the substitutions at locus 2 and locus 3, respectively, but highly dependent on the combination of the substitutions at locus 2 and locus 3. Thus, the SNV at locus 1 is difficult to be detected if $p$ is fixed to 1, but is easy to be detected if there is no restriction on $p$. The existing algorithms V-Phaser and V-phaser2[15,16] were designed to identify minor variants from short-read sequencing data and only leveraged dependence between substitutions at two loci to avoid combinatorial explosion. This is equivalent to fixing $p$ to 1, and making these algorithms unable to detect the SNVs in Supplementary Fig. 1. The proposed RSM algorithm has no restriction on $p$ and can avoid combinatorial explosion.

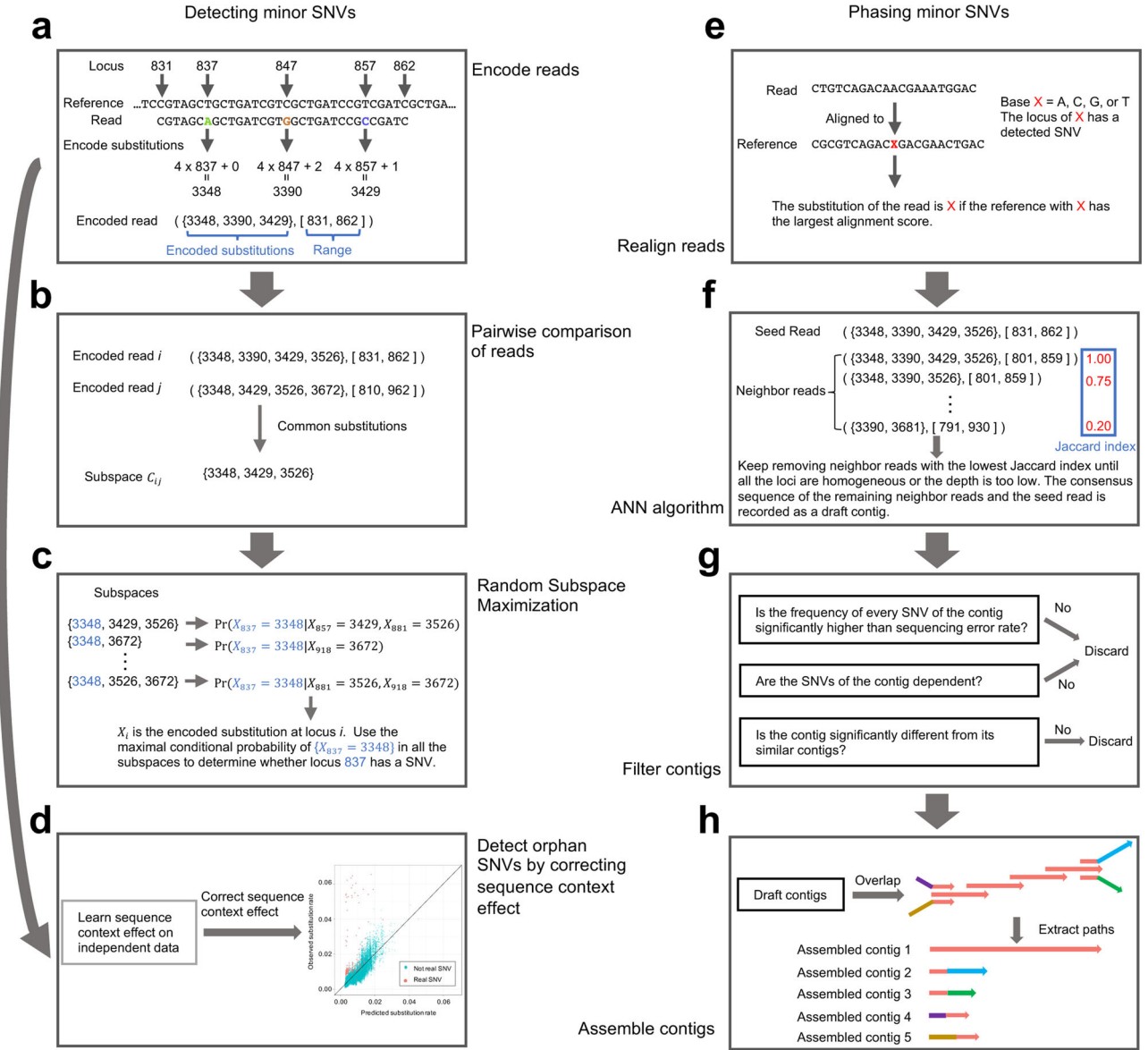

**Fig. 2 The main steps of iGDA.** Details are in the "Methods" section. **a** Encoding reads by using a single integer to represent both locus and identity of each substitution. **b** Generating subspaces by pairwise comparison of reads. **c** An illustrative example of the Random Subspace Maximization algorithm (RSM). **d** Detecting orphan SNVs by correcting sequence-context effect learnt from independent data. **e** Realigning each read to reduce reference bias. **f** An illustrative example of the Adaptive-Nearest Neighbor clustering algorithm (ANN). **g** Filtering contigs by frequencies and correlations of SNVs and similarities between contigs. **h** Assembling filtered contigs by overlap graph.

If a SNV is the only SNV in the genome, we call it an orphan SNV. The proposed framework that uses the conditional substitution rate to detect SNVs cannot detect orphan SNVs because its basic assumption is that there are multiple real SNVs in the same genome. We propose a single-locus-based algorithm to overcome this limitation (Fig. 2d). We discovered that the substitution error rate is very different from locus to locus and it is highly predictable by sequence context (Fig. 3). We trained a gradient boosting model[30] on independent public data and predicted the substitution error rate for each locus. We then adopted a likelihood ratio test to compare the observed substitution rate to the predicted substitution error rate and reported a SNV if they are significantly different (details are in "Methods").

**Phasing minor SNVs.** Intuitively, the reads of the same genome should be clustered together and the consensus sequence of each cluster can be used to phase minor SNVs. Herein, we propose an algorithm called ANN to cluster the reads and the consensus sequence of each cluster is called a draft contig (Fig. 2e, f) (details are in "Methods"). To reduce noise, loci with no detected SNVs are masked before applying ANN algorithm. A major advantage of ANN algorithm is that it can estimate the number of clusters automatically while clustering the reads. To reduce the false-positive rate of the draft contigs, we adopted a two-step filter to remove unreliable draft contigs (Fig. 2g). Intuitively, the SNVs in the same draft contig should be dependent with each other and the difference between two similar draft contigs should be statistically significant.

The lengths of the draft contigs are usually smaller than genome size. To maximize the range where the minor SNVs can be phased, we assemble the draft contigs using an algorithm inspired by overlap graph[31] (Fig. 2h) (details are in "Methods"'). The assembled draft contigs are called contigs.

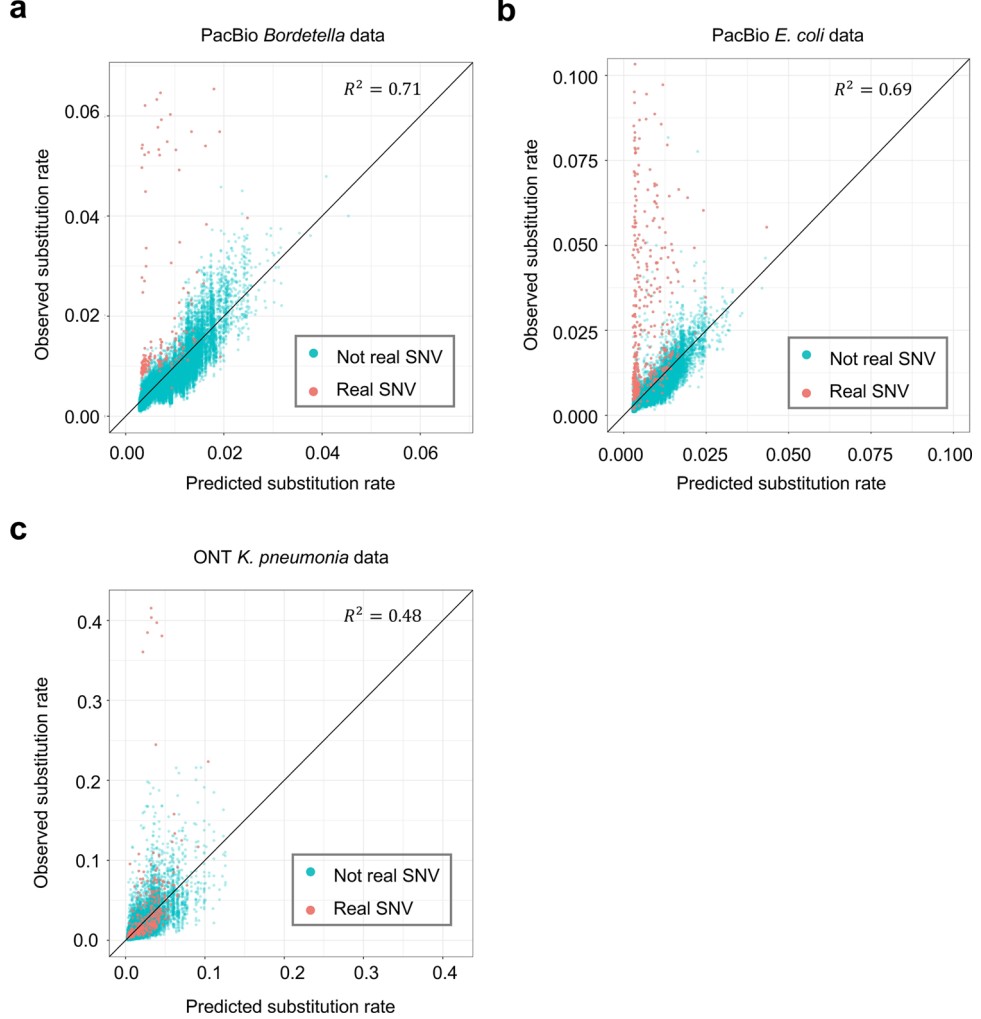

**Fig. 3 Predicting substitution error rate by a sequence-context-effect model trained on independent data.** **a** Prediction of substitution error rate on the PacBio *Bordetella* spp. data. **b** Prediction of substitution error rate on the PacBio *E. coli* data. The axis range is set to [0,0.1], and the data points out of the range are not shown. **c** Prediction of substitution error rate on the ONT *K. pneumoniae* data with DNA methylation masked. Source data are provided as a Source Data file.

**Evaluating performance on pooled PacBio sequencing data.** We constructed two datasets to test the accuracy of iGDA. The first dataset is a mixture of PacBio sequencing data of 186 *Bordetella* spp. samples, and the second dataset is a mixture of 155 *Escherichia coli* samples. The datasets have been previously published and their accession IDs in the SRA database (https://www.ncbi.nlm.nih.gov/sra) are listed in Supplementary Data file 1. The average sequencing depths of pooled data are 29,208× for *Bordetella* spp. and 19,175× for *E. coli*. We downloaded the raw data in HDF format from SRA, and filtered the reads by requiring the estimated read quality (r.q.) greater than 0.75. The estimated r.q. were extracted from the native HDF file. Bases with quality value (QV) less than a threshold were masked. We tested four thresholds, 0, 8, 10, and 12, respectively. We aligned the filtered reads to the reference genomes of *Bordetella pertussis* Tohama I (NCBI Reference Sequence ID is NC_002929.2) for the *Bordetella* spp. data and *Escherichia coli* K12 MG1655 (NCBI reference sequence ID is NC_000913.3) for the *Escherichia coli* data by minimap2[32], respectively. To minimize the alignment ambiguity caused by the aligner, we realigned the reads mapped to the negative strand by aligning their reverse complementary sequences. We only retained the reads aligned to the concatenated *rpoB* and *rpoC* region, which is highly conserved. The

1-based coordinates of the reference genomes is [11662, 20018] for *B. pertussis* Tohama I and [4181245, 4189573] for *E. coli* K12 MG1655. We pooled the realigned reads aligned to the concatenated *rpoB* and *rpoC* region for *Bordetella* spp. and *E. coli* respectively to construct the two datasets. To evaluate the accuracy of iGDA, we ran PacBio's genome consensus module (https://github.com/pacificbiosciences/genomicconsensus) on the aligned reads of each sample with default parameters to obtain the consensus genome sequences and SNVs. The union of the SNVs was used as a benchmark to evaluate the accuracy of detecting SNVs. The genome sequence of an individual sample is defined as a real contig and was used to evaluate the accuracy of contigs reported by iGDA. We merged samples (real contigs) with identical SNV profiles and calculated the relative abundances of the merged samples by the ratio between the number of reads aligned to each sample and the total number of aligned reads. The relative abundances of the samples distinct from the reference genome range from 0.25% to 3.05% for the *Bordetella* spp. data, and range from 0.30% to 1.92% for the *E. coli* data. The average relative abundances are 0.82% and 0.74% for the *Bordetella* spp. data and the *E. coli* data, respectively.

For detecting minor SNVs, we tested three algorithms—a single-locus method (SL), which simply uses the substitution rate

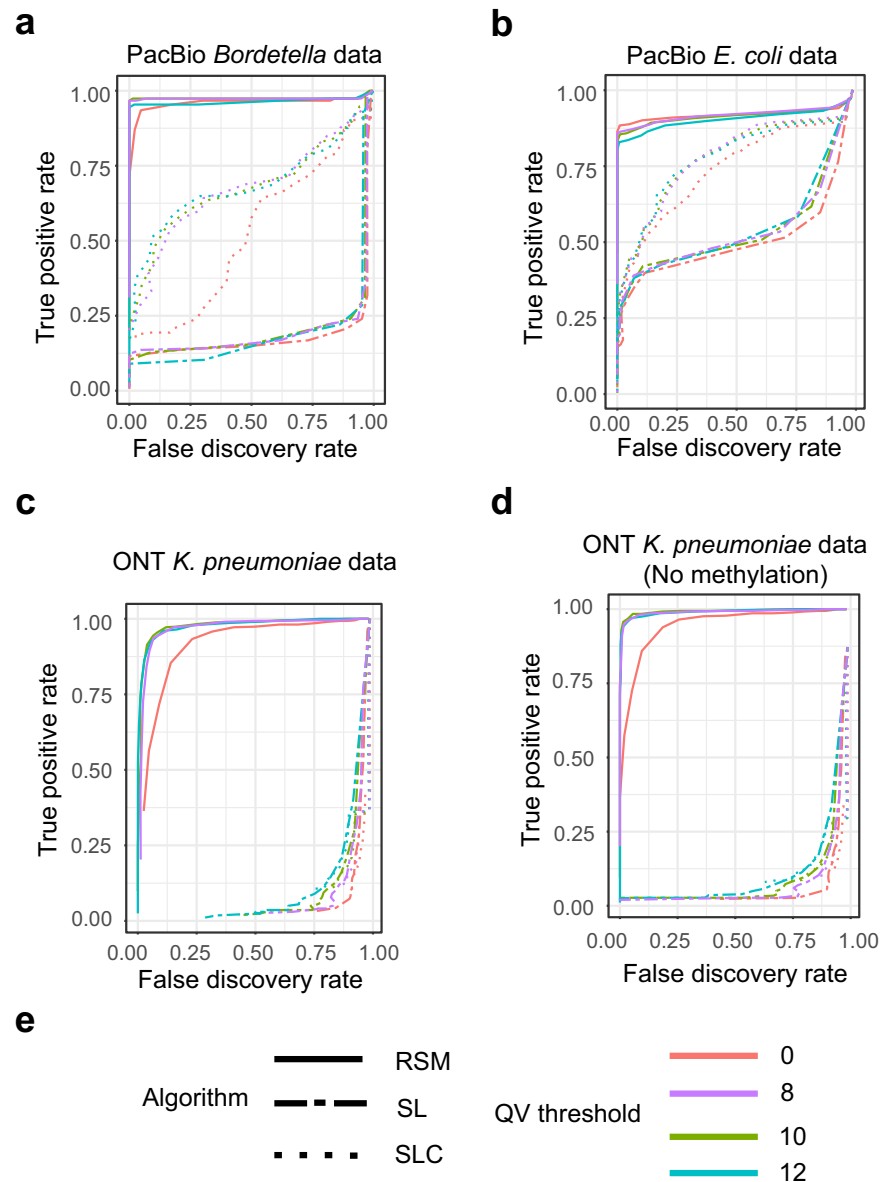

**Fig. 4 The accuracy of detecting minor SNVs on pooled sequencing data. a** The accuracy on PacBio *Bordetella* spp. data. **b** The accuracy on PacBio *E. coli* data. **c** The accuracy on ONT *K. pneumoniae* data. **d** The accuracy on ONT *K. pneumoniae* data with DNA methylation masked. **e** The legend of subfigures **a**–**d** RSM Random Subspace Maximization algorithm, SL single-locus algorithm, SLC single-locus algorithm with correcting sequence-context effect, and QV quality value. True positive rate = number of correctly detected SNVs/number of real SNVs. False discovery rate = 1 − number of correctly detected SNVs/number of detected SNVs. Source data are provided as a Source Data file.

of each locus to detect SNVs; a context-aware single-locus method (SLC), which uses the substitution rate of each locus with correcting sequence-context effect (details are in "Methods"); and the proposed RSM algorithm—on these two test datasets. The results indicate that RSM algorithm greatly outperforms the two single-locus methods and achieves a high accuracy (Fig. 4a, b). With masking bases with QV lower than 8, iGDA detected 96.7% and 85.8% of the real SNVs at a FDR lower than 1% for the *Bordetella* spp. data and *E. coli* data, respectively. Besides, correcting the sequence-context effect substantially increases the detection accuracy of the single-locus methods. The threshold of base QV also has a minor impact on the accuracy. A non-zero threshold increases the accuracy on the *Bordetella* spp. data (Fig. 4a), but decreases the accuracy on the *E. coli* data (Fig. 4b). This might be because masking bases with low QV removes some sequencing errors but reduces effective sequencing depth. We also

evaluated the accuracy of RSM algorithm under different SNV frequencies (Supplementary Fig. 2a, b) and different sequencing depths (Supplementary Fig. 3a, b). The results show that RSM algorithm can accurately detect minor SNVs even for those with frequencies lower than 1%.

For phasing minor SNVs, we evaluated the ANN algorithm on these two datasets, where the bases with QV less than 8 were masked. The phasing accuracy is evaluated by the accuracy of each assembled contig, which is defined as the Jaccard index[33] with its closest real contig. The average accuracies of the assembled contigs are 98.9% and 98.3% for the *Bordetella* spp. data and *E. coli* data, respectively (Fig. 5a). Jaccard index between an iGDA-inferred contig and a real contig is the ratio between the number of shared SNVs and the total number of unique SNVs in their overlapped region. The IGV (Integrative Genomics Viewer)[34] snapshot of the contigs obtained from the *Bordetella*

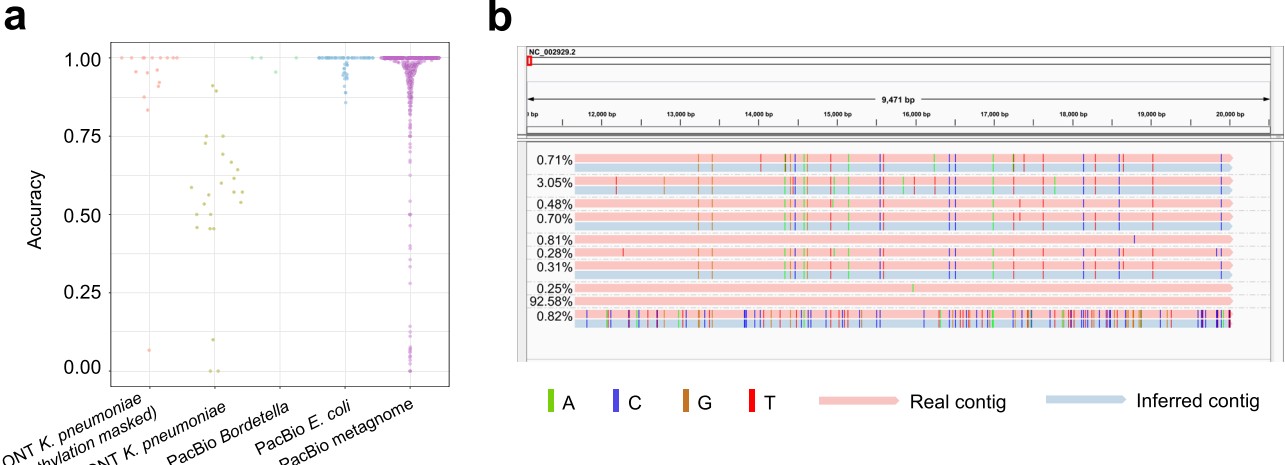

**Fig. 5 The accuracy of phasing minor SNVs. a** The sina plot of accuracy of phasing minor SNVs on the four testing datasets. **b** The IGV snapshot of the contigs inferred by iGDA on the PacBio *Bordetella* spp. data. An inferred contig is grouped with its most similar real contig (measured by Jaccard index). Relative abundance is shown to the left of each contig. Source data are provided as a Source Data file.

spp. data and the *E. coli* data are shown in Fig. 5b and Supplementary Fig. 4. The results show that the iGDA-inferred contigs match the real contigs very well, even for the real contigs with frequencies lower than 1%. In Fig. 5b, there are five real contigs that are not detected by our algorithm. One of them has no SNV (the reference genome); two of them only have a single orphan SNV with very low frequency, which is hard for the RSM algorithm to detect; and two of them are highly similar to another genome. The results indicate that the minor SNVs can be phased effectively except for the genomes that have an orphan SNV or are highly similar to another genome.

**Evaluating performance on pooled ONT sequencing data**. We tested iGDA on a dataset consisting of a mixture of ONT sequencing data of 65 *Klebsiella pneumoniae* samples. The SRA IDs are listed in Supplementary Data file 2. We downloaded the raw data in fastq format from the SRA database (https://www.ncbi.nlm.nih.gov/sra), filtered, and trimmed the reads using fastp[35]. The reads with average QV less than 8 were discarded, and the first 50 bp and the last 200 bp were trimmed for each read. Similar to the PacBio data, we used four thresholds, 0, 8, 10, and 12, respectively, to mask bases with low QV. The reads were then aligned to the reference genome of *K. pneumoniae* subsp. pneumoniae HS11286 (NCBI reference sequence ID is NC_016845.1). We realigned the reads mapped to the negative strand by aligning their reverse complementary sequences. We only retained the reads aligned to the concatenated *rpoB* and *rpoC* region, whose one-based coordinate is [227354, 235682]. We then pooled the aligned reads to construct the testing data. To evaluate the accuracy of iGDA, we downloaded assembly for each sample in the pooled data (Supplementary Data file 2) from NCBI (https://www.ncbi.nlm.nih.gov/assembly) and aligned the assembled genomes to the reference genome using MUMmer[36]. The union of the SNVs reported by MUMmer was used as a benchmark to evaluate the accuracy of detecting SNVs. The genome sequence of an individual sample is defined as a real contig and was used to evaluate the accuracy of contigs reported by iGDA. We used the same method in the previous section to merge identical samples and obtain the relative abundance of each sample. The relative abundances range from 0.20% to 9.30%, and the average relative abundance is 3.20%.

Due to the unique sequencing mechanism of ONT, DNA methylation can affect the raw sequencing signal and substantially increase the base-calling error rate of methylated bases (Supplementary Fig. 5). The base caller used in the public ONT data in this study is Albacore (version 2.0) (https://github.com/Albacore/albacore). To avoid the impact of DNA methylation, we developed an algorithm to identify DNA methylation motifs in bacteria without using raw signal of ONT data (details are in Methods). We masked loci within five bases to the DNA methylation motifs before applying iGDA to this dataset.

The result shows that the RSM algorithm substantially outperforms the single-locus methods to detect minor SNVs, and achieves a high accuracy (Fig. 4c). With DNA methylation and bases with QV lower than 10 masked, iGDA detected 92.8% of the real SNVs at FDR lower than 1%. With masking no DNA methylation but masking bases with QV lower than 10, iGDA detected 41.3% of the real SNVs at FDR lower than 1%. Thus, masking DNA methylation increases the accuracy of the RSM algorithm (Fig. 4d), which demonstrates the importance of removing DNA methylation or applying a methylation-aware base caller to detecting minor SNVs from ONT data. Masking bases with low QV can substantially increase the accuracy and different thresholds have similar accuracies (Fig. 4c, d). In contrast to PacBio data, correcting sequence context does not significantly increase the detection accuracy of the single-locus methods. We speculate that this is because the prediction power of sequence context on the ONT data is weaker than that on the PacBio data (Fig. 3). We also evaluated the accuracy of RSM algorithm under different SNV frequencies (Supplementary Fig. 2c, d) and different sequencing depths (Supplementary Fig. 3c, d). The results show that RSM algorithm can accurately detect minor SNVs even for those with frequencies lower than 1%.

DNA methylation has a large impact on the accuracy of phasing minor SNVs. With masking loci affected by methylation and bases with QV lower than 10, the average accuracy of assembled contigs is 91.0% (Fig. 5a). However, without masking loci affected by methylation, the average accuracy of assembled contigs is only 54.5% (Fig. 5a). An IGV snapshot of methylation-masked contigs is shown in Supplementary Fig. 6. The result shows that the iGDA-inferred contigs match the real contigs very well with DNA methylation masked. It is critical to reduce the

impact of DNA methylation by whole-genome amplification (WGA) or by adopting a methylation-aware base caller.

**De novo identification of multiple Borrelia burgdorferi strains from long-read metagenomic data.** To test whether iGDA can be applied to identify multiple strains of the same species from metagenomic data, we constructed a metagenomic dataset by mixing PacBio sequencing data of 11 *B. burgdorferi* strains, the causal agent of Lyme disease[37], and 744 other bacterial samples. The SRA IDs, species, and strains are in Supplementary Data file 3. We filtered the reads by requiring r.q. value greater than 0.75. r.q. was extracted from the native HDF files. Bases with QV less than 8 were masked. We then aligned the reads to the reference genome of *B. burgdorferi* B31 (NCBI reference sequence ID is NC_001318.1), and realigned the reverse complementary of the reads mapped to the negative strand. To evaluate the accuracy of iGDA, we assembled genome of each *B. burgdorferi* strain using flye[38] and aligned the assembly to the reference genome using MUMmer[36] to obtain benchmark SNVs.

We ran iGDA on the realigned data and constructed 753 contigs. The average accuracy of the contigs is 93.5% (Fig. 5a) and contig length is up to 139 kb. The IGV snapshots of the contigs reported by iGDA show that multiple strains of *B. burgdorferi* can be clearly identified by iGDA (Fig. 6a, Supplementary Figs. 7 and 8). The minimal divergence of a region where the *B. burgdorferi* strains can be distinguished is 0.011% (details are in "Methods"'"). To further evaluate the accuracy of iGDA, we performed MLST (Multilocus Sequence Typing)[39] on the contigs and the genome sequence of each strain using the database at https://pubmlst.org/borrelia (details are in "Methods"). In MLST, we aligned iGDA-inferred contigs and the genome sequence of each strain to the MLST database, consisting of known alleles of the eight house-keeping genes in *Borrelia* spp., to find the best matches. The result shows that most of the alleles that present in the genome sequence of each strain can be found in the iGDA-inferred contigs, and there is no false positive alleles (Fig. 6b). The alleles of the adjacent house-keeping genes, *pyrG*, *recG*, *clpX*, and *pepX*, can be phased by the contigs reported by iGDA (Fig. 6b).

It is worth to note that some genome regions in Fig. 6a are not covered by any contig. We call these regions missed regions and call the SNVs not covered by any contig missed SNVs. We found that there are usually multiple strains that are highly similar to each other in the missed region. In the example shown in Supplementary Fig. 7, at least four samples have highly similar sequences in the missed region. Some missed regions have no SNV compared to the reference genome because iGDA does not report contigs with no SNV. In the example in Supplementary Fig. 8, samples SRR7967871 and SRR7967873 have several large missed regions, which have no SNV compared to the reference genome. To further assess the impact of highly similar strains on the performance of iGDA, we calculated Jaccard index of SNVs for each pair of the *B. burgdorferi* samples, and found that some samples are highly similar to each other. The result in Supplementary Fig. 10a indicates that samples SRR7967879, SRR7967880, SRR7967872, SRR7968340, and SRR7968341 are highly similar to each other, and sample SRR7967869 is highly similar to sample SRR7968342. We constructed a new dataset where only one sample is retained out of the highly similar strains. Specifically, we excluded samples SRR7967879, SRR7967880, SRR7967872, SRR7968340, and SRR7968342 from the samples listed in Supplementary Data file 3, and reran iGDA on the new data. The result shows that the accuracy of each contig is not significantly changed by excluding highly similar strains (Supplementary Fig. 10b). However, the length of contigs and proportion of SNVs covered by contigs are substantially

increased (Supplementary Figs. 10c, d, and 9). The species other than *B. burgdorferi* have limited impact on the results because most of the reads from these species (Supplementary Data file 3) cannot be aligned to the reference genome of *B. burgdorferi*, and 99.93% of the aligned reads are aligned to 16S ribosomal RNA or 23S ribosomal RNA.

We tested iGDA on the mimic metagenomic data using a CentOS Linux machine with 96-core 2.70 GHz Intel 8168 CPU and 1 Tb RAM. It took 51 min using 32 threads (3.6 CPU hours) to detect SNVs and took 1.5 h using 32 threads (5.5 CPU hours) to phase SNVs. The peak memory was 5.3 Gb.

## Discussion

We here present iGDA, an open-source tool implementing several innovative algorithms that can achieve a high accuracy for detecting and phasing minor SNVs. iGDA makes it feasible to study a number of previously challenging problems, such as constructing strain-level genome sequence in microbiome samples and identifying genome sequence of pathogens in samples with co-infection. The RSM and ANN algorithms proposed in this work are generic methods and can be extended to apply to single-cell genome sequencing data or 10X genomics linked-read[40] data. In addition to genome sequencing, these algorithms have the potential to be applied in RNA sequencing data as well. For example, with an alternative prepossessing procedure, these algorithms can be used to decipher the heterogeneity of A-to-I RNA editing using long-read sequencing.

A major limitation of iGDA is that its high accuracy relies on the presence of multiple SNVs. Therefore, iGDA has reduced accuracy to detect orphan SNVs with very low frequency. Besides, the presence of highly similar genomes will reduce the accuracy of iGDA.

DNA methylation can induce correlated substitution errors on ONT data and reduce the accuracy of iGDA. Masking DNA methylation can increase the accuracy of iGDA on ONT data. Using WGA to remove DNA methylation is a solution to this issue. Another solution is to use a base caller that can correct methylation-induced error, but there is no such tool currently available according to our best knowledge.

In this work, we only detect minor SNVs because they are less affected by alignment ambiguity compared to insertions and deletions (Indel). Alignment ambiguity means an Indel might be located to multiple loci in the genome but the corresponding alignment scores are equal. To extend our RSM and ANN algorithms to detect minor Indels or other more complicated variants, an alternative way to represent variants and alignments is needed.

## Methods

**Leveraging multiple loci to detect SNVs.** For the $i$th aligned read, we encode its substitution at locus $k$ of the reference genome by the following formula:

$$s_{ik} = \begin{cases} 4k & r_{ik} \neq t_k, r_{ik} = A \\ 4k+1 & r_{ik} \neq t_k, r_{ik} = C \\ 4k+2 & r_{ik} \neq t_k, r_{ik} = G, \\ 4k+3 & r_{ik} \neq t_k, r_{ik} = T \\ \epsilon & r_{ik} = t_k \end{cases} \quad (1)$$

where $r_{ik}$ is the base (short for nitrogenous base) of the $i$th aligned read at locus $k$, $t_k$ is the base at locus $k$ of the reference genome and $\epsilon$ is an empty element, which is formally defined by $\{\epsilon\} = \emptyset$. The first locus of the reference genome is 0 throughout this paper unless otherwise stated. The $i$th read is represented as a set of substitutions and its covering range (Fig. 2a) and is denoted by

$$R_i = (S_i, [b_i, e_i]). \quad (2)$$

$b_i$ and $e_i$ are the start and end loci of the region covered by the read, respectively, and $S_i$ is

$$S_i = \{s_{ib_i}, s_{ib_i+1}, ..., s_{ie_i}\}. \quad (3)$$

The most intuitive way to detect SNVs is to use the substitution rate of each locus. Formally, we denote the encoded substitution at locus $k$ as a random variable

**a**

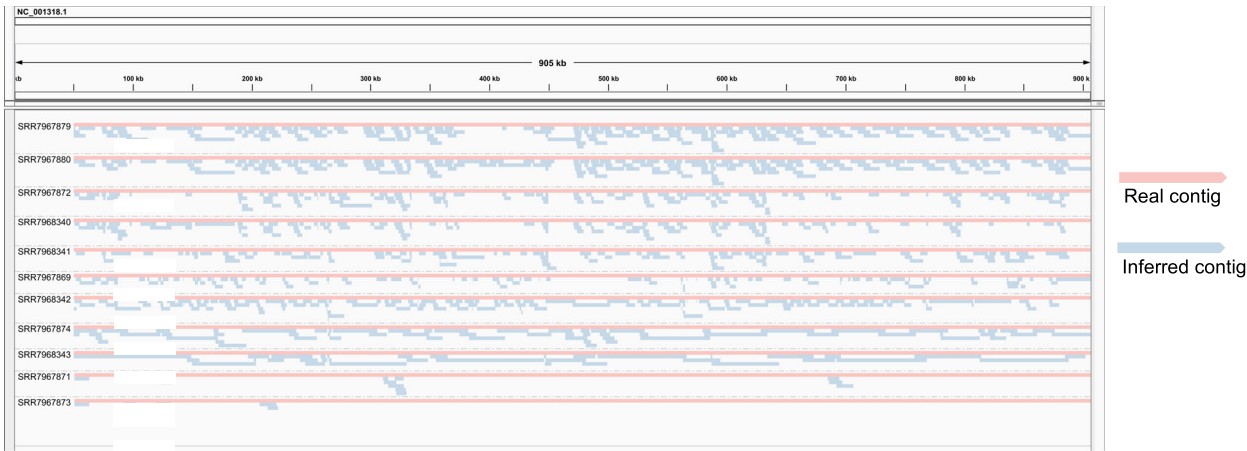

**b**

| Distance | | 297 kb | 120 kb | 90 kb | 8 kb | 41 kb | 18 kb | 233 kb |
|---|---|---|---|---|---|---|---|---|
| | **nifS** | **clpA** | **rplB** | **pyrG** | **recG** | **clpX** | **pepX** | **uvrA** |
| SRR7967879 | nifS_11 | clpA_14 | rplB_1 | pyrG_1 | recG_11 | clpX_1 | pepX_1 | uvrA_10 |
| SRR7967880 | nifS_11 | clpA_14 | rplB_1 | pyrG_1 | recG_11 | clpX_1 | pepX_1 | uvrA_10 |
| SRR7967872 | nifS_11 | clpA_14 | rplB_1 (99.84%) | pyrG_1 | recG_1 | clpX_1 | pepX_1 | uvrA_10 |
| SRR7968340 | nifS_11 | clpA_14 | rplB_1 | pyrG_1 | recG_1 | clpX_1 | pepX_119 | uvrA_10 |
| SRR7968341 | nifS_11 | clpA_14 | rplB_1 | pyrG_1 | recG_1 | clpX_1 | pepX_1 | uvrA_157 |
| SRR7967869 | nifS_12 | clpA_15 | rplB_8 | pyrG_1 | recG_11 | clpX_9 | pepX_8 | uvrA_16 |
| SRR7968342 | nifS_12 | clpA_15 | rplB_8 | pyrG_1 | recG_11 | clpX_9 | pepX_8 | uvrA_16 |
| SRR7967874 | nifS_5 | clpA_6 | rplB_1 | pyrG_1 | recG_7 | clpX_1 | pepX_1 | uvrA_8 |
| SRR7968343 | nifS_4 | clpA_10 | rplB_1 | pyrG_1 | recG_6 | clpX_5 (99.84%) | pepX_6 | uvrA_6 |
| SRR7967871 | nifS_1 | clpA_1 | rplB_1 | pyrG_1 | recG_1 | clpX_1 | pepX_1 | uvrA_1 |
| SRR7967873 | nifS_1 | clpA_1 | rplB_1 | pyrG_1 | recG_1 | clpX_1 | pepX_1 | uvrA_1 |

Alleles phased    Alleles phased

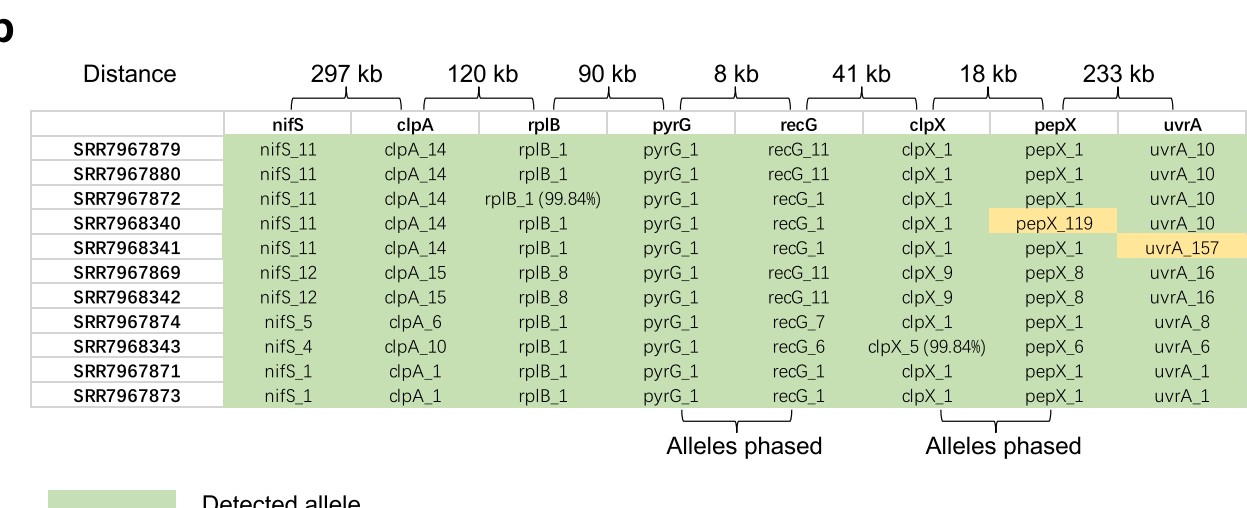

Detected allele

Missed allele

**Fig. 6 *De novo* identification of multiple *Borrelia burgdorferi* strains from PacBio metagenomic data. a** The IGV snapshot of the contigs inferred by iGDA from the metagenomic data. Each contig is grouped with its closest real contig (*B. burgdorferi* strain). **b** Multilocus Sequence Typing (MLST) of *B. burgdorferi* in the metagenomic data. The columns are the alleles of the eight house-keeping genes used in MLST. Each row is the alleles of the genome of each sample (strain). The row names are the accession numbers of each sample in Sequence Read Archive (SRA, https://www.ncbi.nlm.nih.gov/sra). An allele is detected if it matches a contig inferred by iGDA. There are two alleles that have no 100% match in the MLST database, and their similarities to the closest alleles in the database are shown in the brackets. All the other alleles have a 100% match in the database. Source data are provided as a Source Data file.

$X_k$, and denote probability of the event $\{X_k = x_k\}$ as $Pr(X_k = x_k)$, where $x_k \in \{4k, 4k + 1, 4k + 2, 4k + 3\}$. Substitution rate is defined as the estimated $Pr(X_k = x_k)$, which is

$$\hat{Pr}(X_k = x_k) = \frac{|\{i \mid x_k \in S_i\}|}{|\{i \mid k \in [b_i, e_i]\}|},\qquad(4)$$

where $\{\cdot\}$ is a set and $|\cdot|$ is the number of elements in a set. Intuitively, in Eq. (4), the numerator is the number of reads with substitution $x_k$ at locus $k$, and the denominator is the number of reads covering locus $k$. Due to the high error rate of long-read sequencing data, it is inaccurate to detect minor variants using substitution rate alone (Fig. 1b). Herein, we leverage the information of multiple loci to increase the detection accuracy. Assuming sequencing errors are independent with each other, real SNVs are likely to be present if there are multiple reads containing the same set of substitutions (Fig. 1a). The conditional probability of $\{X_k = x_k\}$ given other real SNVs of the same genome is therefore much larger than the marginal probability of $\{X_k = x_k\}$ if $x_k$ is a real SNV, because these real SNVs are positively dependent (Fig. 1a, c). Formally, the conditional probability of event $\{X_k = x_k\}$ given $p$ other substitutions is defined as

$Pr(X_k = x_k | X_{g_1} = x_{g_1}, X_{g_2} = x_{g_2}, ..., X_{g_p} = x_{g_p})$, which is estimated by

$$\hat{Pr}(X_k = x_k | X_{g_1} = x_{g_1}, X_{g_2} = x_{g_2}, ..., X_{g_p} = x_{g_p}) = \frac{|\{i \mid \{x_k, x_{g_1}, x_{g_2}, ..., x_{g_p}\} \subseteq S_i\}|}{|\{i \mid \{x_{g_1}, x_{g_2}, ..., x_{g_p}\} \subseteq S_i, k \in [b_i, e_i]\}|}.$$

(5)

Intuitively, in Eq. (5), the numerator is the number of reads containing substitution $x_k$ and the $p$ other substitutions, and the denominator is the number of reads that contain the $p$ other substitutions and cover locus k. The $p$ loci, $g_1, g_2, ...,$ and $g_p$ are called dependent loci. As $x_{g_1}, x_{g_2}, ..., x_{g_p}$, and $p$ in Eq. (5) are unknown, the estimated maximal conditional probability of event $\{X_k = x_k\}$ given $p$ other substitutions is used to detect SNVs and is formally defined by

$$H(x_k) = \max_{p, x_{g_1}, x_{g_2}, ..., x_{g_p}} \{\hat{Pr}(X_k = x_k | X_{g_1} = x_{g_1}, X_{g_2} = x_{g_2}, ..., X_{g_p} = x_{g_p})\}.\qquad(6)$$

The substitution $x_k$ is detected as a real SNV if $H(x_k)$ is larger than a threshold (0.65 in this study). $H(x_k)$ is also called maximal conditional substitution rate. To avoid high variance of the estimated $Pr(X_k = x_k | X_{g_1} = x_{g_1}, X_{g_2} = x_{g_2}, ..., X_{g_p} = x_{g_p})$ (Eq. 5), we require that $|\{i \mid \{x_{g_1}, x_{g_2}, ..., x_{g_p}\} \subseteq S_i, k \in [b_i, e_i]\}| >= v_{\min}$, and $v_{\min} = 25$ in this study. Sequencing errors at multiple loci that are very close to each

other might induce slightly dependent substitutions. To avoid the impact of dependent substitutions induced by sequencing errors, we require that locus $k$ and loci $g_1, g_2, ..., g_p$ are not too close. Specifically, we require $HD(k, g_s) \geq 15$ for any $g_s \in \{g_1, g_2, ..., g_p\}$. $HD(k, g_s)$ is the homopolymer distance between locus $k$ and locus $g_s$, and is defined as the number of homopolymers between the two loci. A homopolymer is a set of consecutive identical bases, and a base with no identical adjacent bases is also defined as a special homopolymer with size equal to 1.

It is computationally infeasible to enumerate all combinations of $p$ loci to estimate $H(x_k)$ in Eq. (6). It is important to note that it is insufficient to detect SNVs accurately by restricting the number of dependent loci $p$ to a certain number. In the example shown in Supplementary Fig. 1, $H(x_k)$ fails to detect the real SNVs if $p$ is restricted to 1. Likewise, we can also have similar examples if $p$ is restricted to another number greater than 1. In this work, we developed an algorithm called RSM that can estimate $H(x_k)$ efficiently without restricting $p$.

**Detecting SNVs by RSM algorithm**

*The greedy algorithm and its theoretical accuracy.* We introduce a fast but inaccurate greedy algorithm to estimate $H(x_k)$ (Eq. 6), and then improve its accuracy by RSM in the next section. To estimate $H(x_k)$ for substitution $x_k$ at locus $k$, we only need to consider dependent loci in range $[k - t_l, k + t_r]$, where

$$t_l = \max_t \{ |\{i \mid [k-t, k] \subseteq [b_i, e_i]\}| > 0 \}$$

$$t_r = \max_t \{ |\{i \mid [k, k+t] \subseteq [b_i, e_i]\}| > 0 \}.$$

$[b_i, e_i]$ is the covering range of read $R_i$ (equation (2)), and $|\{\cdot\}|$ is the number of elements in set $\{\cdot\}$. Intuitive, $[k - t_l, k + t_r]$ is the largest range where $[k-t_l, k]$ and $[k, k + t_r]$ are fully covered by at least one read. We estimate $Pr(X_k = x_k|X_g = x_g)$ by Eq. (5) for each locus $g \in [k - t_l, k + t_r] \cap \{k\}^c$ ($\{\cdot\}^c$ is complement of a set), and sort the loci according to $Pr(X_k = x_k|X_g = x_g)$ in descending order. The sorted loci are denoted as $\{s_1, s_2, ..., s_{t_l+t_r}\}$, and $Pr(X_k = x_k|X_{s_{t-1}} = x_{s_{t-1}}) \geq Pr(X_k = x_k|X_{s_t} = x_{s_t})$. We keep adding locus $s_t$ to $\{s_1, s_2, ..., s_{t-1}\}$ if $\hat{Pr}(X_k = x_k|X_{s_1} = x_{s_1}, X_{s_2} = x_{s_2}, ..., X_{s_t} = x_{s_t}) > \hat{Pr}(X_k = x_k|X_{s_1} = x_{s_1}, X_{s_2} = x_{s_2}, ..., X_{s_{t-1}} = x_{s_{t-1}})$ and stop if otherwise. $\hat{Pr}(X_k = x_k|X_{s_1} = x_{s_1}, X_{s_2} = x_{s_2}, ..., X_{s_v} = x_{s_v})$ based on the final $v$ selected loci $\{s_1, s_2, ..., s_v\}$ is used to estimate $H(x_k)$.

The naive greedy algorithm described above avoids combinatorial explosion but might have low accuracy. We assume $x_k, x_{g'_1}, x_{g'_2}, ..., x_{g'_p}$ are $p + 1$ real SNVs of the same genome, and $x_{g'_1}, x_{g'_2}, ..., x_{g'_p}$ are the only $p$ substitutions that can maximize $\hat{Pr}(X_k = x_k|X_{g_1} = x_{g_1}, X_{g_2} = x_{g_2}, ..., X_{g_p} = x_{g_p})$. Formally, $H(x_k) = \hat{Pr}(X_k = x_k|X_{g'_1} = x_{g'_1}, X_{g'_2} = x_{g'_2}, ..., X_{g'_p} = x_{g'_p})$, and $\hat{Pr}(X_k = x_k|X_{g_1} = x_{g_1}, X_{g_2} = x_{g_2}, ..., X_{g_p} = x_{g_p}) < \hat{Pr}(X_k = x_k|X_{g'_1} = x_{g'_1}, X_{g'_2} = x_{g'_2}, ..., X_{g'_p} = x_{g'_p})$ if $\{g_1, g_2, .., g_p\} \neq \{g'_1, g'_2, ..., g'_p\}$. Assuming $k, g'_1, g'_2, ..., g'_p$ are the only loci with real SNVs in $[k - t_l, k + t_r]$, we define signal-to-noise ratio by

$$\rho_0 = Pr(\hat{Pr}(X_k = x_k|X_{g_s} = x_{g_s}) > \max_{x_{g_t}}\{\hat{Pr}(X_k = x_k|X_{g_t} = x_{g_t})\}),$$

where $x_{g_s} \in \{x_{g'_1}, x_{g'_2}, .., x_{g'_p}\}$ and $g_t \notin \{g'_1, g'_2, .., g'_p\}$. $g_t \notin \{g'_1, g'_2, .., g'_p\}$ is equivalent to $g_t \in [k - t_l, k + t_r] \cap \{k, g'_1, g'_2, ..., g'_p\}^c$. For any locus $g_s \in \{g'_1, g'_2, .., g'_p\}$, the probability that it is selected by the greedy algorithm is denoted as $Pr(g_s \in \{s_1, s_2, ..., s_v\})$, where $\{s_1, s_2, ..., s_v\}$ is the $v$ loci selected by the greedy algorithm. Without loss of generality, assuming $v \leq p$ and sequencing errors are independent,

$$Pr(g_s \in \{s_1, s_2, ..., s_v\}) \leq Pr(g_s \in \{s_1, s_2, ..., s_p\})$$
$$= \prod_{g_t \notin \{g'_1, g'_2, ..., g'_p\}} Pr(\hat{Pr}(X_k = x_k|X_{g_s} = x_{g_s}) > \max_{x_{g_t}}\{\hat{Pr}(X_k = x_k|X_{g_t} = x_{g_t})\})$$
$$= \rho_0^{(t_l + t_r - p)}.$$

The probability that the greedy algorithm correctly estimates $H(x_k)$ is

$$Pr(H(x_k) = \hat{Pr}(X_k = x_k|X_{s_1} = x_{s_1}, X_{s_2} = x_{s_2}, ..., X_{s_v} = x_{s_v})) = Pr(\{g'_1, g'_2, ..., g'_p\} \subseteq \{s_1, s_2, ..., s_v\})$$
$$\leq Pr(g_s \in \{s_1, s_2, ..., s_v\})$$
$$= \rho_0^{(t_l + t_r - p)}.$$
(7)

According to inequation (7), assuming $t_l \geq 2000$, $t_r \geq 2000$, and $p = 1$, which is a typical setting for long-read sequencing data, the probability that the greedy algorithm correctly estimates $H(x_k)$ is less than $3.5 \times 10^{-18}$ even if $\rho_0 = 0.99$. The key factor leading to the failure of the greedy algorithm is selecting from too many loci ($t_l + t_r$ loci). We propose an algorithm called RSM to reduce the number of loci to be considered in the next section.

*Improving accuracy of the greedy algorithm by RSM.* First, we measure the similarity between two reads, $R_i$ and $R_j$, by a modified Jaccard index[33], which is defined by

$$Jaccard(R_i, R_j) = \frac{|S_i \cap S_j|}{|(S_i \cup S_j) \cap [4\max(b_i, b_j), 4\min(e_i, e_j) + 3]|}$$
(8)

where $Jaccard(R_i, R_j) = 0$ if the denominator is 0. We require

$$|[\max(b_i, b_j), \min(e_i, e_j)]| \geq l_{\min}$$

where $l_{\min}$ is the minimal length of the overlap region between the two compared reads. We used $l_{\min} = 0.5(e_i - b_i)$ in this work. Intuitively, the Jaccard index between two reads is the ratio between number of common substitutions shared by the two reads and the total number of substitutions of the two reads in their overlapped region. Then, for a read $R_i$, we select $w$ most similar reads according to the Jaccard index. For each read $R_j$ in these $w$ selected reads, we generate a set of substitutions shared by $R_i$ and $R_j$. Formally, $C_{ij} = S_i \cap S_j$. $C_{ij}$ is called a subspace (Fig. 2b), and we can generate $w \times m$ subspaces if there are $m$ reads. We used $w = 100$ in this work. For a substitution $X_k \in C_{ij}$, we estimate its maximal conditional probability of $\{X_k = x_k\}$ in subspace $C_{ij}$, which is defined by

$$H_{C_{ij}}(x_k) = \max_{\{x_{g_1}, x_{g_2}, ..., x_{g_p}\} \subseteq C_{ij}} \left\{ \hat{Pr}(X_k = x_k|X_{g_1} = x_{g_1}, X_{g_2} = x_{g_2}, ..., X_{g_p} = x_{g_p}) \right\}$$
$$= \max_{x_{g_1}, x_{g_2}, ..., x_{g_p}} \left\{ \frac{|\{t \mid \{x_k, x_{g_1}, x_{g_2}, ..., x_{g_p}\} \subseteq (S_t \cap C_{ij})\}|}{|\{t \mid \{x_{g_1}, x_{g_2}, ..., x_{g_p}\} \subseteq (S_t \cap C_{ij}), k \in [b_t, e_t]\}|} \right\},$$
(9)

using the greedy algorithm described in the previous section by only considering the substitutions in $C_{ij}$. Thus, compared to the original greedy algorithm, the number of loci to be considered is substantially reduced. We then use

$$\hat{H}(x_k) = \max_{C_{ij}}(\hat{H}_{C_{ij}}(x_k))$$
(10)

to estimate the maximal conditional probability of $\{X_k = x_k\}$ defined by Eq. (6). $\hat{H}_{C_{ij}}(x_k)$ is the maximal conditional probability of $\{X_k = x_k\}$ in subspace $C_{ij}$ estimated by the greedy algorithm. The whole procedure of estimating $H(x_k)$ in the $w \times m$ subspaces is called RSM (Fig. 2c).

*Theoretical accuracy of RSM algorithm.* Without the loss of generality, we denote $\{x'_{g_1}, x'_{g_2}, ..., x'_{g_p}\}$ as the only set of substitutions that maximizes $\hat{Pr}(X_k = x_k|X_{g_1} = x_{g_1}, X_{g_2} = x_{g_2}, ..., X_{g_p} = x_{g_p})$, and $\Omega$ as the set of subspaces containing $\{x_k, x'_{g_1}, x'_{g_2}, ..., x'_{g_p}\}$. For a subspace $C_t \in \Omega$, the probability that the greedy algorithm finds $\{x'_{g_1}, x'_{g_2}, ..., x'_{g_p}\}$ is denoted as $Pr(\hat{H}_{C_t}(x_k) = H(x_k))$, where $H_k$ is defined by Eq. (6). The probability that RSM algorithm finds $\{x'_{g_1}, x'_{g_2}, ..., x'_{g_p}\}$ is

$$Pr(\hat{H}(x_k) = H(x_k)) = Pr(\cup_{C_t \in \Omega}\{\hat{H}_{C_t}(x_k) = H(x_k)\})$$
$$= 1 - Pr(\cap_{C_t \in \Omega}\{\hat{H}_{C_t}(x_k) \neq H(x_k)\}).$$
(11)

Assuming $Pr(\hat{H}_{C_t}(x_k) = H(x_k)) > 0$, and according to the chain rule of joint probability,

$$Pr(\cap_{C_t \in \Omega}\{\hat{H}_{C_t}(x_k) \neq H(x_k)\}) =$$
$$Pr(\hat{H}_{C_1}(x_k) \neq H(x_k)) \prod_{t=2}^{|\Omega|} Pr(\hat{H}_{C_t}(x_k) \neq H(x_k)|\hat{H}_{C_{t-1}}(x_k) \neq H(x_k), ..., \hat{H}_{C_1}(x_k) \neq H(x_k)),$$

where $Pr(\hat{H}_{C_t}(x_k) \neq H(x_k)|\hat{H}_{C_{t-1}}(x_k) \neq H(x_k), ..., \hat{H}_{C_1}(x_k) \neq H(x_k)) < 1$ if $C_t \notin \{C_{t-1}, C_{t-2}, ..., C_1\}$. As sequencing depth increases, $|\Omega|$ increases, and $Pr(\cap_{C_t \in \Omega}\{\hat{H}_{C_t}(x_k) \neq H(x_k)\})$ converges to 0. Thus, $Pr(\hat{H}(x_k) = H(x_k))$ (Eq. (11)) converges to 1 as sequencing depth increases. Intuitively, with infinite sequencing depth, RSM algorithm is guaranteed to detect real SNVs correctly if these SNVs have larger maximal conditional probabilities than sequencing errors.

*Detecting orphan SNVs by correcting sequence-context effect.* As RSM algorithm requires multiple real SNVs, it cannot detect orphan SNVs. An orphan SNV is the only SNV of the genome. We have to rely on the single-locus algorithm described in Eq. (4) to detect orphan SNVs. However, the substitution rate of a locus is not only affected by real SNVs but also affected by the sequence context of the locus. We built a gradient boosting[30] model to learn the sequence-context effect and corrected it by the following likelihood ratio method (Fig. 2d). For a substitution $x_k$ at locus $k$, its likelihood ratio is

$$LR(x_k) = \frac{Binomial(t_k; n_k, p_1)}{Binomial(t_k; n_k, p_0)},$$
(12)

where $Binomial(x; n, p)$ is the probability mass function of binomial distribution with parameters $n$ and $p$, and

$$t_k = |\{i \mid x_k \in S_i\}|$$
$$n_k = |\{i \mid k \in [b_i, e_i]\}|$$
$$p_1 = \frac{t_k}{n_k}$$

$p_0 =$ Predicted sequencing error rate by sequence context

The substitution $x_k$ is detected as a SNV if $LR(x_k)$ is larger than a threshold. We used a threshold of 50 in this work. Calculation of $p_0$ is introduced in the next

section. To reduce FDR, we also required a detected SNV has a substitution rate higher than 0.1 for PacBio data and 0.2 for ONT data respectively.

*Modeling sequence-context effect on sequencing error rate.* Error rate of long-read sequencing is strongly affected by sequence context (Fig. 3). For locus $i$, we define its one upstream homopolymer and one downstream homopolymer as its sequence context (Supplementary Fig. 11). We adopted the gradient boosting model implemented by xgboost (version 0.90)[30] to predict substitution rate of each locus by its sequence context. For PacBio, we trained the model on a dataset consisting of 79 PacBio RS II runs with P6-C4 chemistry and a dataset consisting of 24 PacBio RS II runs with P4-C2 chemistry, respectively (SRA IDs of the data are listed in Supplementary Data file 4). As the sequence-context effects on these two datasets are highly similar, we only used the model trained on the P6-C4 data for the analysis. For ONT, we trained the model on a dataset consisting of eight MinION runs with R9.4 chemistry (SRA IDs of the data are listed in Supplementary Data file 4). We tuned three parameters in gradient boosting, step size (eta in xgboost), number of trees (num_round in xgboost) and maximal depth of trees (max_depth in xgboost) and used the parameters with the highest fivefold cross-validation accuracy (Supplementary Data file 5). We used $R^2$ as the measurement of accuracy, which is defined by

$$R^2 = \frac{\sum_{i=1}^{n}(y_i - \hat{y}_i)^2}{\sum_{i=1}^{n}(y_i - \bar{y})^2}$$

where $y_i$ is the substitution rate of a sequence context, $\hat{y}_i$ is the predicted substitution rate, $\bar{y}$ is the average substitution rate, and $n$ is the number of unique sequence contexts. For PacBio, step size, number of trees and maximal depth of trees with the highest accuracy are 0.01, 2000, and 10, respectively. For ONT, step size, number of trees, and maximal depth of trees with the highest accuracy are 0.1, 2000, and 10, respectively.

We also masked bases with QV thresholds 8, 10, and 12, and trained three different models on the masked data. Each model is used in the detection algorithm which masks bases with the same QV threshold. In the case of not masking any base, we predicted substitution rate using the trained model on the three pooled sequencing datasets (Fig. 3). The results show that the substitution error rate is strongly affected by sequence context and can be well predicted by our model.

**Phasing minor SNVs.** To detect whether multiple minor SNVs are from the same DNA molecule, we proposed an algorithm called ANN. As the reads inevitably have errors, an intuitive way to phase minor SNVs is to cluster the reads and use the consensus sequences of each cluster to phase the minor SNVs. However, an intrinsic difficulty of clustering algorithms is to determine the number of clusters, which is unknown. The ANN algorithm can directly estimate the number of clusters from data.

*Adaptive-Nearest-Neighbors clustering.* First, we realigned each read to reduce reference bias (details are in the next section) and only retained detected SNVs for each read. Formally, for read $R_i$ (Eq. 2), we use

$$\widetilde{S}_i = S_i \cap \{\text{Detected SNVs}\}, \tag{13}$$

where $S_i$ is defined in Eq. (3).

The intuitive idea of ANN algorithm is that all loci should be homogeneous by piling up the reads in each cluster (Supplementary Fig. 12). A locus is homogeneous if it satisfies the following condition. For locus $k$, its substitution rate satisfies

$$\widetilde{Pr}(X_k = x_k) = \frac{|\{i \mid x_k \in \widetilde{S}_i\}|}{\sum_{d=0}^{3} |\{i \mid 4k + d \in \widetilde{S}_i\}| + |\{i \mid r_{ik} = t_k\}|} \in [0, p_{\text{lim}}] \cup [1 - p_{\text{lim}}, 1], \tag{14}$$

where $x_k \in \{4k, 4k+1, 4k+2, 4k+3\}$, $r_{ik}$ is the base of read $i$ at locus $k$, and $t_k$ is the base of the reference genome at locus $k$. In this work, We set $p_{\text{lim}} = 0.2$ for the PacBio data and $p_{\text{lim}} = 0.3$ for the ONT data. Intuitively, the numerator is the number of substitutions with alternative base equals to $x_k$ at locus $k$, the first term of the denominator is the total number of substitutions, and second term of the denominator is the number of bases equal to the reference genome at locus $k$. Locus $k$ is homogeneous if $\widetilde{Pr}(X_k = x_k) \leq p_{\text{lim}}$ or $\widetilde{Pr}(X_k = x_k) \geq 1 - p_{\text{lim}}$ for any $x_k \in \{4k, 4k+1, 4k+2, 4k+3\}$.

For a read $i$ (called seed read), we sorted its $q$ most similar reads according to the Jaccard index (Eq. 8), and kept discarding the most dissimilar one until all loci covered by the seed read are homogeneous or maximal coverage of the loci is smaller than a threshold (10 in this work) (Fig. 2f). We recorded the consensus sequence as a draft contig if all the loci are homogeneous (Supplementary Fig. 12). We calculated the Jaccard index of each read with all the draft contigs, and assigned the read to the contig with the largest Jaccard index. A read is assigned to the reference genome if its largest Jaccard index is smaller than 0.5. The abundance of a contig is defined as the number of reads assigned to it.

A problem of the algorithm described above is that the alignment is affected by reference bias and homogeneous loci could be mistaken for heterogeneous loci. Reference bias is the phenomenon that the substitution rate of a real SNV at a homogeneous locus is significantly lower than $1 -$ substitution error rate (Supplementary Fig. 13a).

*Reference bias and local realignment.* For each detected SNV, we adopted standard Smith–Waterman algorithm implemented by SeqAn (version 2.4) (https://www.seqan.de) to realign reads to four modified reference sequences with A, C, G, or T at each locus with a detected SNV. The scores of match, mismatch, gap open, and gap extension are 2, −4, −4, and −2, respectively, and the score of a base aligned to base N or a masked low-QV base is 0. To avoid the high computational cost, we only realigned 21 homopolymers whose center is the locus with detected SNV. For each read, the modified base in the reference sequence with the highest alignment score is recorded as a substitution of the read (Fig. 2e and Supplementary Fig. 14). We tested the realignment method on a single *E. coli* dataset (SRA ID is ERS718594), which is presumably homogeneous. The result shows that local realignment can substantially reduce reference bias (Supplementary Fig. 13b). The average substitution rate of loci with real SNVs is 84.8% before realignment, and the average substitution rate of loci with real SNV is 95.9% after realignment. We performed local realignment before ANN algorithm in our analysis.

*Filtering draft contigs.* To reduce false-positive rate of the inferred draft contigs by ANN algorithm, we adopted a two-step algorithm to filter the draft contigs (Fig. 2g). In the first step, we tested whether the frequency of each individual SNV in each contig is significantly higher than the sequencing error rate and whether SNVs in each contig are independent using the Bayes factor. The contig is filtered if the frequency of any of its SNVs is not significant and its SNVs are independent (Supplementary Fig. 15a). In the second step, we compared the contigs pairwise, and the contig with lower abundance in each pair is filtered if the contigs are not significantly different according to the Bayes factor (Supplementary Fig. 15b).

*Assembling draft contigs.* The length of the draft contigs obtained by ANN algorithm is usually smaller than genome size, except in a few cases like a virus genome. Therefore, we have to assemble the draft contigs to obtain the whole picture of the underlined genomes in the sequenced sample. We borrowed the idea of overlap graph[31] from de novo genome assembly to assemble the draft contigs. We denoted each draft contig as a vertex in a graph and compared the contigs pairwise. For a draft contig $i$, we linked it to another draft contig $j$ by adding an edge from vertex $i$ to vertex $j$ if all the three criteria are met: (1) the two draft contigs are identical in their overlapped region; (2) the number of overlapped SNVs is more than 50% of the number of SNVs in contig $i$ or that in contig $j$, or the length of overlapped region is more than 50% of the length of contig $i$ or that of contig $j$; (3) the genome coordinate of the end locus of contig $i$ is smaller than that of contig $j$. We then removed redundant edges by transitive reduction[41] (Supplementary Fig. 16a, b). A contig is constructed by concatenating draft contigs, which are in an unambiguous path. A path is an unambiguous path if the three criteria are met: (1) in-degree of the start vertex is not 1; (2) out-degree of the end vertex is not 1 or a daughter vertex of the end vertex has more than one parental vertices; (3) in-degrees and out-degrees of the vertices other than the start vertex and the end vertex are 1 (Fig. 2h and Supplementary Fig. 16c). We then filtered the contigs using the two-step filter introduced in the previous section. We calculated the Jaccard index of each read to all the contigs, and assigned the read to the contig with the largest Jaccard index. A read is assigned to the reference genome if its largest Jaccard index is smaller than 0.5.

Errors in contigs might make the contigs from the same strain disconnected in the overlap graph. To overcome this limitation, we firstly removed contained contig if its Jaccard index is higher than 0.9 with the overlapped region of a longer contig fully covering it, and then connected contigs if the Jaccard index in their overlapped region is higher than 0.9.

**Detecting bacterial methylation motifs from ONT data without raw signal.** As the raw-signal files of ONT data are usually huge and not publicly available, we developed an algorithm to detect DNA methylation motifs without raw signal. For each individual ONT data file before pooling, we extracted the flanking sequences (40 bp long) of loci whose substitution rates are greater than 0.15, and detected motifs in the flanking sequences using the motif caller developed by PacBio (https://github.com/PacificBiosciences/MotifMaker)[42]. We only retained the motifs that match the known bacterial methylation motifs in REBASE (http://rebase.neb.com/rebase/rebase_methylase_recseqs.txt)[43]. Thus, our methylation-motif detection algorithm is conservative and only detects known motifs. We only discovered two known motifs, CCWGG and CGCATC, on the ONT data. W represents A or T.

***Borrelia* MLST.** We downloaded the allele sequences of the eight house-keeping genes from https://pubmlst.org/bigsdb?db=pubmlst_borrelia_seqdef&page=downloadAlleles, and aligned them to the iGDA-inferred contigs and the genome sequence of each *Borrelia burgdorferi* strain using MUMmer[36]. If a contig or genome sequence has no 100% match in the allele database, we reported the allele with the highest percent identity in the MUMmer output.

**Evaluating the minimal divergence that two conspecific strains can be distinguished**. We only retained the iGDA-reported contigs that is 100% identical to a true genome sequence and only has an unique closest true genome sequence. These retained contigs can be used to distinguish conspecific strains. We calculated the divergence between two contigs by

$$\text{Divergence}(\text{contig1, contig 2}) = \frac{\text{number of different SNVs}}{\text{length of overlapped region}}$$

**Software version and parameter setting**
*Flye (version 2.6-release)*. In the PacBio metagenomic data, we used "flye -t 16 –pacbio-raw -g 2m".

*MUMmer (version 3.1)*. We used "nucmer -c 150 -g 500 -l 12 –maxmatch" for alignment, and "show-snps -l -T -H" to obtain SNVs. To avoid the impact of repeats we used "mummerplot −−filter" before "show-snps -l -T -H" for the metagenomic data.

*minimap2 (version 2.17-r968-dirty)*. We used "minimap2 –secondary=no -ax map-pb" for PacBio data, and "minimap2 –secondary=no -ax map-ont" for ONT data.

*iGDA (version 1.02)*. We used "igda_pipe_detect -s 0 -a 0" to detect minor SNVs and "igda_pipe_phase" with default parameters to phase minor SNVs for the PacBio metagenomic data. We used "igda_pipe_detect -f 0 -s 0 -a 0" and "igda_pipe_phase" for the other PacBio data. We used "igda_pipe_detect -f 0 -m ont -s 0 -a 0" to detect minor SNVs and "igda_pipe_phase -m ont" to phase minor SNVs for the ONT data.

**Reporting summary**. Further information on research design is available in the Nature Research Reporting Summary linked to this article.

## Data availability
All the data used in this work are publicly available at SRA (https://www.ncbi.nlm.nih.gov/sra). The accession IDs are listed in Supplementary Data files 1-4. Source data are provided with this paper.

## Code availability
The source code of iGDA and example data are available at https://github.com/zhixingfeng/iGDA. The DOI of this repository is 10.5281/zenodo.4637922.

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

## Acknowledgements
The project is supported by funds from the Steven & Alexandra Cohen Foundation.

## Author contributions

Z.F. designed and implemented the computational models and algorithms of iGDA. B.W. proposed and tested the methods to improve speed of iGDA. Z.F. performed the data analysis with support from B.W. Z.F. designed the experiments to evaluate iGDA on metagenomic data with the support from J.C. and E.E.S. Z.F. wrote the manuscript with input from all authors.

## Competing interests

E.E.S. is on the scientific advisory board of Pacific Biosciences. The other authors declare no competing interests.
