## [Peer Review File · Nature Communications]

Reviewer #1 (Remarks to the Author):

The authors proposed iGDA, an open-source tool aiming for accurately detecting and phasing minor SNVs with very low frequencies by long-read sequencing data. Below are it has some restrictions.

1. iGDA detects minor SNVs with their frequencies as low as 0.2% but it strongly relies on sequencing depth. Please evaluate the performance of iGDA over different sequencing depths and different frequencies of SNVs.
2. iGDA should apply filter in detecting minor SNVs (VAF or frequency < 0.1). Besides, it should be dependent on sequencing depth and $H(x_k)$. Please evaluate the parameter selection with different sequencing depths and $H(x_k)$. Especially for $H(x_k)$, does it vary with different sequencing platforms and species?
3. In Fig3, the correlation between predicted and observed substitution rate by PacBio higher than by ONT, why? In Fig3C, some substitution rates of real SNVs are larger than 0.2 but they do not appear in the plot of PacBio, why?
4. In Fig4, the authors wanted to show more information to demonstrate the better performance of RSM than other methods but it is very unclear. Please improve Fig4.
5. The authors wanted to show the performance of SNV detection by correcting sequence context effect. Please compare the performance before and after correcting sequence context effect.
6. Minor SNVs phasing is closely correlated with the distance of these SNVs. How is the performance of iGDA with different sequencing lengths and the distance between minor SNVs. Besides, the phasing accuracy is unclear in the paper. More details should be included.

Reviewer #2 (Remarks to the Author):

In their manuscript, Feng et al presented a new method for phasing minor single-nucleotide variants from long-read error-prone sequencing data. Due to their length, long reads are good candidates for phasing, and there is a need for such a method. Also, the approach that includes the conditional substitution rate is plausible. Presented results look promising. However, I have several concerns.

- (1) In the manuscript, authors mentioned that circular consensus sequencing (CSS) Pacbio reads are not suitable for phasing because there are 10-20 fold shorter than continuous long reads (CLR). I argue that the same method presented in this paper can be used for the newest generation hifi reads (Sequel 2 CSS reads) which median size can reach 30 kbp. I deem that overlap method used in this paper might be successfully used for hifi reads. I am aware that due to the shortage of available dataset it is not easy to prepare equivalent datasets for hifi reads, but I'm not at all convinced that they could not be used for phasing due to their length. Recently, several authors (ie. Heng Li's and Adam Phillippy's groups) have showed that using hifi reads is possible to achieve phased genomes with higher contiguity than genomes assembled by either nanopore or CLR Pacbio reads.
- (2) Information about required computation resources is not presented, i.e. there is no running time, required memory, and used equipment.
- (3) I agree with the authors that the error rate of long-reads is affected by sequence context, ie. homopolymeric regions. However, input features for the prediction model are not specified.
- (4) To increase accuracy, the authors masked methylation motifs, masked bases with lower quality values, and ignore indels. It would be worth knowing what the number of SNV was before and after masking.
- (5) The proposed approach might phase just a portion of genomes. It would be very informative if authors provide information about the average and maximum proportion of a strain they might phase.
- (6) Equation 14 should be additionally clarified.

Reviewer #3 (Remarks to the Author):

The authors provided a new tool iGDA to address the problem of SNV calling and clustering for

long reads sequencing. Given many other various sites, they estimated maximal conditional probability of a candidate SNV by Random Subspace Maximization (RSM). To detect orphan SNVs, they modeled sequence context effect on sequencing error rate. Then, a Adaptive Nearest-Neighbors clustering (ANN) algorithm was introduced to estimate the number of clusters in piled up reads. Though HiFi-reads will make the SNV calling a much easier task, but the algorithm advance on analysis raw sequencing data is still highly expected and valued.

iGDA was tested a mimic whole genome metagenomic data. I am confused of the inferred contigs on the Fig 6 and Fig S6, why so many overlapped and contained contigs with identical alleles? Were them inferred draft contigs or unambiguous contigs? If the latter, the assembly of inferred draft contigs was a failure. Besides, I think lots of readers care about the computing efficiency of iGDA on mimic metagenomic data, can the authors provide the run time and peak RAM numbers in main text.

I know that iGDA is not designed for phasing diploid genome assemblies, but in theory it can detect SNVs and clustering long reads along assembled contigs which is concerned by more users. I just wonder whether those algorithms outperform long reads phasing tools, e.g. longshot. A popular hard core method should not focus on a highly specific problem when it already contains the solution of general problem (phasing diploid or polyploid contigs).

REVIEWER COMMENTS

We thank the reviewers for their insightful comments, which are highly valuable for improving the quality of the manuscript. We listed our one-to-one response to the comments below and revised the manuscript accordingly. In the revised manuscript, the text highlighted in yellow is the changed text, and the text highlighted in green is the text mentioned in this letter.

Reviewer #1 (Remarks to the Author):

The authors proposed iGDA, an open-source tool aiming for accurately detecting and phasing minor SNVs with very low frequencies by long-read sequencing data. Below are it has some restrictions.

1. iGDA detects minor SNVs with their frequencies as low as 0.2% but it strongly relies on sequencing depth. Please evaluate the performance of iGDA over different sequencing depths and different frequencies of SNVs.

We agree with the reviewer that the performance of iGDA depends on sequencing depth and SNV frequency. Following the reviewer's suggestion, we have now included the accuracy of iGDA on different SNV frequencies in Figure S2 and evaluated the impact of sequencing depth on the testing data in Figure S3 by subsampling the reads. In general, iGDA has higher detect accuracy for SNVs with higher frequency. The accuracy of iGDA increases as sequencing depth increases. We also revised the main text line 190-192 on page 10 and line 243-245 on page 13 respectively to demonstrate the evaluation result as following:

“We also evaluated accuracy of RSM algorithm under different SNV frequencies (Figure S2A and Figure S2B) and different sequencing depth(Figure S3A and Figure S3B).”

“We also evaluated accuracy of RSM algorithm under different SNV frequencies (Figure S2C and Figure S2D) and different sequencing depth(Figure S3C and Figure S3D).”

2. iGDA should apply filter in detecting minor SNVs (VAF or frequency<0.1). Besides, it should be dependent on sequencing depth and $H(x_k)$. Please evaluate the parameter selection with different sequencing depths and $H(x_k)$. Especially for $H(x_k)$, does it vary with different sequencing platforms and species?

Although designed to detect minor SNVs with low frequency, iGDA is able to detect SNVs with higher frequency as well in the same framework. iGDA reports all the detected SNVs with different frequencies and the end-users can filter SNVs by frequency according to the specific biological problems if needed.

There is only one parameter, v_{min} , in iGDA to detect SNVs. v_{min} , which is the minimal denominator in the formula to calculate conditional substitution rate (equation (5) on page 18),

was used to ensure the variance of the estimated conditional substitution rate is not too high and fixed as 25 in this study (line 358, page 18). This parameter is not dependent on sequencing depth or $H(x_k)$.

$H(x_k)$ is the maximal conditional substitution rate of substitution x_k at locus k given the substitutions at p other loci (equation (6) on page 18), and is the output of RSM algorithm in iGDA. The value of $H(x_k)$ is calculated from input data and does not affect parameter selection. Any differences in the input data, including those caused by different sequencing platforms or composition of species/strains, might lead to different $H(x_k)$.

3. In Fig3, the correlation between predicted and observed substitution rate by PacBio higher than by ONT, why? In Fig3C, some substitution rates of real SNVs are larger than 0.2 but they do not appear in the plot of PacBio, why?

As PacBio and ONT adopt totally different sequencing mechanisms and base calling algorithms, all these factors might affect the correlation between substitution rate and sequence context.

The PacBio *Bordetella* dataset does NOT have any real SNV with a frequency higher than 0.2. The PacBio *E. coli* dataset does have some real SNVs with frequencies higher than 0.2, but we set the axis range to be [0, 0.1] for better visualization of the context effect (R^2 was calculated only using the green points, which are substitution rate of the loci with no real SNV. R^2 is NOT affected by setting the axis range because the data points outside of the axis range were also included to calculate R^2). Without setting the axis range, the scatter plot looks like Figure R1 below, where the correlation between context effect and substitution rate is unclear because most of the data points are in the left corner. We did not set axis range for the ONT data. To avoid misunderstanding of the data characteristics, we have now added a sentence (shown below) to emphasize that we set axis range in the caption of Figure 3B on page 9.

“The axis range is set to [0,0.1], and the data points out of the range are not shown.”

Figure R1. Correlation between predicted substitution rate and observed substitution rate on PacBio *E.coli* data without setting axis range.

4. In Fig4, the authors wanted to show more information to demonstrate the better performance of RSM than other methods but it is very unclear. Please improve Fig4.

We tried to represent the performance of RSM algorithm (RSM) as solid curves in Figure 4. We realized some of these curves might look like dotted curves due to the overlap of the curves with different colors (each color represents the QV cutoff to filter bases with low QV). Following the reviewer's suggestion, we have now corrected this and improved Figure 4 for better visualization.

5. The authors wanted to show the performance of SNV detection by correcting sequence context effect. Please compare the performance before and after correcting sequence context effect.

The accuracies before and after correcting sequence context effect are shown in Figure 4. The dashed curves (figure legend = SL) show the accuracy before correction and the dotted curves (figure legend = SLC) show the accuracy after correction. The definition of the figure legend is in the caption of Figure 4, and the description of the single-locus methods is in line 179 - 182 on page 10. The accuracy is clearly improved after correcting sequence context effect for PacBio data (Figure 4A and 4B) but not for ONT data (Figure 4C and 4D), which we have described in the main text.

“Besides, correcting sequence-context effect substantially increases detection accuracy of the single-locus methods.” (line 186 – 187 page 10)

“In contrast to PacBio data, correcting sequence context does not significantly increase detection accuracy of the single-locus methods. We speculate that this is because the prediction power of sequence context on the ONT data is weaker than that on the PacBio data (Figure 3)” (line 241 – 243 page 13)

6. Minor SNVs phasing is closely correlated with the distance of these SNVs. How is the performance of iGDA with different sequencing lengths and the distance between minor SNVs. Besides, the phasing accuracy is unclear in the paper. More details should be included.

We agree with the reviewer that the performance of iGDA depends on read length and distance between minor SNVs. For SNV detection, the algorithm cannot benefit from leveraging conditional substitution rate if an SNV has no neighboring real SNVs that are covered by the same read. It is difficult to phase SNVs if their distances are so large that only few or no reads cover them.

We realized that phase accuracy was not explicitly defined in the original manuscript. Following the reviewer's suggestion, we have now added the following sentence to the main text.

“The phasing accuracy is evaluated by the accuracy of each assembled contig, which is defined as its Jaccard index with its closest real contig.” (line 195 – 196 page 10)

A draft contig is the consensus sequence of each reads cluster reported by ANN algorithm of iGDA (defined in line 140-141 on page 6). Assembled contig is the assembly of the draft contigs using overlap graph (Figure 2H) and the genome sequence of an individual sample before mixing is defined as a real contig (line 171-172 page 8). The testing data we used are mixtures of reads from multiple public sequenced samples, so we could obtain the genome sequence of each individual sample by downloading from public databases or *de novo* assembly of each sample. Jaccard index between an assembled contig and a real contig is defined as the ratio between the number of shared SNVs and the total number of unique SNVs in their overlapped region (line 198-199 page 10).

Besides, we also reported N50 of contig length and proportion of the real SNVs that can be covered by the contigs (Figure S10) for each strain. These results demonstrate the distance among SNVs that can be phased and the proportion of SNVs that can be phased.

Reviewer #2 (Remarks to the Author):

In their manuscript, Feng et al presented a new method for phasing minor single-nucleotide variants from long-read error-prone sequencing data. Due to their length, long reads are good candidates for phasing, and there is a need for such a method. Also, the approach that includes the conditional substitution rate is plausible. Presented results look promising. However, I have several concerns.

We are pleased that Reviewer #2 agreed our work presented promising results.

(1) In the manuscript, authors mentioned that circular consensus sequencing (CSS) Pacbio reads are not suitable for phasing because there are 10-20 fold shorter than continuous long reads (CLR). I argue that the same method presented in this paper can be used for the newest generation hifi reads (Sequel 2 CSS reads) which median size can reach 30 kbp. I deem that overlap method used in this paper might be successfully used for hifi reads. I am aware that due to the shortage of available dataset it is not easy to prepare equivalent datasets for hifi reads, but I'm not at all convinced that they could not be used for phasing due to their length. Recently, several authors (ie. Heng Li's and Adam Phillippy's groups) have showed that using hifi reads is possible to achieve phased genomes with higher contiguity than genomes assembled by either nanopore or CLR Pacbio reads.

We agree with the reviewer that it is possible to apply iGDA to HiFi data and HiFi reads might have better phasing contiguity because of the higher read-level accuracy on some data. In the original manuscript, we were trying to emphasize that there is a trade-off between read length and read-level accuracy in CCS. We also noticed that our original statement "However, CCS reduces read length by 10 to 20 fold to achieve low error rates, and read length is critical to phasing minor SNVs" might be interpreted as we imply that CCS is not suitable for phasing, which is inaccurate. We have now deleted this sentence in the revised manuscript. In addition to read length and read accuracy, contiguity of phasing is also affected by algorithm, genome and

sequencing depth, *etc.* So It is difficult to conclude that whether CCS is better for phasing or not in general even with evidence that CCS is better for phasing on a number of genomes using algorithms specifically designed for CCS reads.

(2) Information about required computation resources is not presented, i.e. there is no running time, required memory, and used equipment.

We have added run time, RAM usage, and testing hardware/software environment in line 299 – 302 on page 16.

“We tested iGDA on the mimic metagenomic data using a CentOS Linux machine with 96-core 2.70GHz Intel 8168 CPU and 1 Tb RAM. It took 51 minutes using 32 threads (3.6 CPU hours) to detect SNVs and took 1.5 hours using 32 threads (5.5 CPU hours) to phase SNVs. The peak memory was 5.3 Gb.”

(3) I agree with the authors that the error rate of long-reads is affected by sequence context, ie. homopolymeric regions. However, input features for the prediction model are not specified.

We apologize for the confusion. We used the identity and the number of bases in the neighboring homopolymers of a locus as the input feature. For example, the input feature of TTA~~A~~AACCC is (2, T, 4, A, 3, C), and the response is the substitution rate of the locus ~~A~~. We have now included the input features in the caption of Figure S11.

“The features of the sequence context are the identity and number of bases in these homopolymers. The feature vector of sequence context, TTAA~~A~~ACCC, is (2, T, 4, A, 3, C).”

(4) To increase accuracy, the authors masked methylation motifs, masked bases with lower quality values, and ignore indels. It would be worth knowing what the number of SNV was before and after masking.

We are not sure if the SNV mentioned by the reviewer is the real SNVs or the SNVs detected by iGDA. For real SNVs, they are not affected by masking bases with low QV (quality value) or ignoring indels because these operations only affect the encoded reads, and the real SNVs were obtained by aligning the genome sequence of each strain, which is known in our testing data, to the reference. For the ONT data, 55 out of 433 real SNVs were masked by removing methylation motifs. We did not perform methylation motif masking for PacBio data because methylation has a limited impact on base calling according to its sequencing mechanism.

The impact of masking bases with low QV on the number of detected SNVs is illustrated in Figure 4. The number of detected SNVs with the FDR ≤ 0.01 is listed in the following table. Without masking methylation motifs and bases with low QV, no SNV is detected in the ONT data under FDR ≤ 0.01 . This is possible because the dataset is so noisy that some loci with no SNV might have a very strong signal possibly due to high sequencing errors induced by methylation.

	No QV masking	QV >= 8	QV >= 10	QV >= 12	# of Real SNVs
PacBio Bordetella	137	149	149	147	154
PacBio E. coli	409	393	385	383	458
ONT K. pneumonia (methylation masked)	139	347	353	348	378
ONT K. pneumonia	0	89	180	304	433

(5) The proposed approach might phase just a portion of genomes. It would be very informative if authors provide information about the average and maximum proportion of a strain they might phase.

We showed the distribution of contig length as well as median values and N50 in Figure S10C. The proportion of phased SNVs of each strain is in Figure S10D. The N50 of contig length on the mimic metagenomic data with 11 *Borrelia burgdorferi* strains is 12.0 kb. We also noticed that the existence of highly similar strains leads to fragmented contigs. By removing 5 highly similar strains, the N50 of contig length was increased to 35.8 kb (Figure S10C). The maximum length that can be phased was described in line 266 - 267 on page 14. We used contig length to evaluate the length of the region that can be phased. We could phase up to about 139 kb (longest contig), and this is about 15% of the chromosome size of *Borrelia burgdorferi*.

“The average accuracy of the contigs is 93.5% (Figure 5A) and contig length is up to 139 kb.” (line 266 – 267 page 14)

(6) Equation 14 should be additionally clarified.

We realized that there was a typo in equation 14. The second term of the denominator was missed in the original manuscript. We have now corrected equation 14 and added an intuitive explanation of equation 14 in line 486 – 489 on page 24. We thank the reviewer for pointing this out.

Reviewer #3 (Remarks to the Author):

The authors provided a new tool iGDA to address the problem of SNV calling and clustering for long reads sequencing. Given many other various sites, they estimated maximal conditional probability of a candidate SNV by Random Subspace Maximization (RSM). To detect orphan SNVs, they modeled sequence context effect on sequencing error rate. Then, a Adaptive Nearest-Neighbors clustering (ANN) algorithm was introduced to estimate the number of clusters in piled up reads. Though HiFi-reads will make the SNV calling a much easier task, but the algorithm advance on analysis raw sequencing data is still highly expected and valued.

iGDA was tested a mimic whole genome metagenomic data. I am confused of the inferred contigs on the Fig 6 and Fig S6, why so many overlapped and contained contigs with identical alleles? Were they inferred draft contigs or unambiguous contigs? If the latter, the assembly of inferred draft contigs was a failure. Besides, I think lots of readers care about the computing efficiency of iGDA on mimic metagenomic data, can the authors provide the run time and peak RAM numbers in main text.

The contigs in these figures are assembled contigs corresponding to the unambiguous paths in the overlap graph (Figure S16). However, **we argue that the existence of overlapped contigs with identical alleles does NOT mean the failure of draft contigs assembly.** The overlapped contigs are caused by the branches in the overlap graph (after transitive reduction), and iGDA reports the overlapped contigs in the branches as they are, rather than merging them to form longer contigs. **This is to avoid chimeric errors that incorrectly merge contigs from different strains.** For example, in Figure R2A below (equals to the upper part of Figure S7), contig 1 and contig 2 have identical overlapped regions with contig 3 respectively, but the overlapped regions between contig 1 and contig 2 are different. iGDA reports all of contig 1, 2, and 3 as they are because there is a branch in the overlap graph (Figure 2RB). In this example, contig 1 and contig 3 are from two different strains (different SNVs between these two strains are highlighted with red boxes in Figure R2A), and merging them to form a longer contig will lead to a chimeric error (See Figure R2C below). It is also worth noting that cutting off the overlapped regions will lead to information loss (See Figure R2C below) because the original contigs provide information that the non-overlapping part of each contig is from the same strain with the overlapping part. Thus, to avoid chimeric errors and information loss, contigs in branches of the overlap graph should be reported as they are.

Many of the overlapped contigs are induced by the highly similar strains because they cause a lot of branches in the overlap graph. We already tested iGDA on the mimic metagenomic data by removing 5 highly similar strains as described in line 285 – 293 on page 16 and obtained much fewer overlapped contigs. We have now included the IGV snapshot on the metagenomic data with 5 highly similar strains removed in Figure S9 of the revised manuscript to demonstrate the impact of highly similar strains on phasing SNVs.

The apparently contained contigs are caused by minor errors in the contigs. In the original iGDA algorithms (version 0.9.3) used to generate these results, a contig is discarded as “contained contig” if it is identical to the overlapped region of another longer contig that fully covers it. We admit that this is too stringent, so we discard a contained contig if its similarity with the overlapped region of the longer contig is higher than a threshold (Jaccard index ≥ 0.9) in the latest version of iGDA. Besides, minor errors in the contigs make the draft contigs from the same strain disconnected in the overlap graph because the original assembly algorithm in iGDA requires two draft contigs to have identical overlapped region to be connected. Similar to the contained contigs, we have now updated the overlap algorithm to connect two contigs in the overlap graph by requiring the Jaccard index in their overlapped region higher than 0.9 instead of requiring identical overlapped region. We have updated Figure 6, Figure S4, Figure S6 - S8 using the revised algorithms (Figure S6 in the original manuscript is now Figure S8 in the revised

manuscript). Figure S4 and S6 (IGV snapshot of assembled contigs on the *rpoBC* region of PacBio *E. coli* and ONT *K. pneumoniae* data) only have minor differences from the original figure. Figure 6, S7, and S8 (IGV snapshot of the assembled contigs on the mimic metagenome data) have fewer assembled contigs than the original ones. We have also added the following sentence to the main text.

“Errors in contigs might make the contigs from the same strain disconnected in the overlap graph. To overcome this limitation, we firstly removed contained contig if its Jaccard index is higher than 0.9 with the overlapped region of a longer contig fully covering it, and then connected contigs if the Jaccard index in their overlapped region is higher than 0.9.” (Line 541 – 544 page 26)

Figure R2. The example demonstrating overlapped contigs with identical alleles in branches should be reported as they are. **A**, IGV snapshot of the contigs reported by iGDA. The pink bars are the real contigs obtained from the

known genome sequence of each individual strain (unknown in practice), and the blue bars are the assembled contigs reported by iGDA. The blue bars are grouped with their closest real contig. **B**, The overlap graph of contig 1, 2, and 3. **C**, Three different ways to report contig 1, 2, and 3. iGDA adopts the third way in the bottom.

Following the reviewer's suggestion, we have added information on computational resources including run time, peak RAM, hardware /software environment in line 299 – 302 on page 16 and released the testing data and code at GitHub (<https://github.com/zhixingfeng/iGDA>).

“We tested iGDA on the mimic metagenomic data using a CentOS Linux machine with 96-core 2.70GHz Intel 8168 CPU and 1 Tb RAM. It took 51 minutes using 32 threads (3.6 CPU hours) to detect SNVs and took 1.5 hours using 32 threads (5.5 CPU hours) to phase SNVs. The peak memory was 5.3 Gb.”

I know that iGDA is not designed for phasing diploid genome assemblies, but in theory it can detect SNVs and clustering long reads along assembled contigs which is concerned by more users. I just wonder whether those algorithms outperform long reads phasing tools, e.g. longshot. A popular hard core method should not focus on a highly specific problem when it already contains the solution of general problem (phasing diploid or polyploid contigs).

We agree with the reviewer that iGDA can be applied to phase haplotypes of diploid or polyploid genomes. We compared iGDA and longshot on human data to detect SNVs and phase diploid genome. In our testing dataset (PacBio sequencing data of chromosome 10 of sample NA24385 from Genome in a Bottle), the two algorithms have similar accuracies on detecting SNVs (F-score: iGDA = 0.981, longshot = 0.990) while longshot can phase longer haplotypes (haplotype length N50: longshot = 393 kb vs iGDA = 182 kb). Making explicit assumptions on the number of haplotypes is helpful to improve the performance of phasing algorithms when the assumption holds.

Reviewer #1 (Remarks to the Author):

The authors have addressed my concerns

Reviewer #2 (Remarks to the Author):

The authors answered most of my comments.

I appreciate the authors' intention for mathematical rigour. Yet, sometimes it is difficult to follow. Therefore, some expressions should be clarified

line 376 Formulas for T_I and T_r

lines 409-412, it is not clear what are m and w

equation 14 description - $\Pr \sim (X_k = x_k) \leq \text{plim}$ or $\Pr \sim (X_k = x_k) \geq \text{plim}$ for any x_k ... Why did the authors choose such a definition? It looks that plim can be any value

Minor comments:

- lines 96-97 "multiple sequencing errors are unlikely to repeatedly occur together on the same read". Do the authors mean here on position or read ?

- Figure 2E "if the the reference" two the

- There are no parameters for `minimap2` in Parameter setting section

Reviewer #3 (Remarks to the Author):

The authors had answered my questions.

REVIEWERS' COMMENTS

Reviewer #1 (Remarks to the Author):

The authors have addressed my concerns

Reviewer #2 (Remarks to the Author):

We thank the reviewer for the detailed comments. We revised the manuscript accordingly and responded the comments below.

The authors answered most of my comments.

I appreciate the authors' intention for mathematical rigour. Yet, sometimes it is difficult to follow. Therefore, some expressions should be clarified

line 376 Formulas for T_l and T_r

We added an intuitive explanation of t_l and t_r . We also explicitly defined the notation $\{ \cdot \}$ to make the formal definition of t_l and t_r easier to understand for a broad range of readers.

lines 409-412, it is not clear what are m and w

Both of m and w are variables, and their values are determined by users or the input data.

m is the total number reads in the input data. We believe the sentence "...if there are m reads." implies the definition of m .

w is the number of reads used to construct subspaces for a read R_i . We used an empirically determined value, 100, in this work. A larger w means we will construct more subspaces. In theory, RSM will have better sensitivity with larger w (see equation (11) for theoretical accuracy, number of subspaces is denoted as $|\Omega|$), but use more computational resources. A w larger than 100 has very small impact on the accuracy of RSM in our testing data, so we fixed it as 100.

equation 14 description - $\Pr \sim (X_k = x_k) \leq p_{lim}$ or $\Pr \sim (X_k = x_k) \geq p_{lim}$ for any x_k ... Why did the authors choose such a definition? It looks that p_{lim} can be any value

It is a shame that there is still a typo here. It should be $\widetilde{Pr}(X_k = x_k) \leq p_{lim}$ or $\widetilde{Pr}(X_k = x_k) \geq 1 - p_{lim}$. We corrected this in equation (14).

Minor comments:

- lines 96-97 "multiple sequencing errors are unlikely to repeatedly occur together on the same read". Do the authors mean here on position or read ?

We rephrased this to “the same combination of sequencing errors at multiple loci is unlikely to repeatedly occur together on multiple reads” to make the expression more clear.

- Figure 2E "if the the reference" two the

We corrected it in the revised manuscript.

- There are no parameters for minimap2 in Parameter setting section

We added the version and parameters for minimap2.

We also added the version and parameters for our own software, iGDA, in the section.

Reviewer #3 (Remarks to the Author):

The authors had answered my questions.